# Extracellular traps from activated vascular smooth muscle cells drive the progression of atherosclerosis

Ming Zhai[1,5], Shiyu Gong[1,5], Peipei Luan[1], Yefei Shi[1], Wenxin Kou[1], Yanxi Zeng[1], Jiayun Shi[1], Guanye Yu[1], Jiayun Hou[2], Qing Yu[1], Weixia Jian[3], Jianhui Zhuang[1], Mark W. Feinberg [4] & Wenhui Peng [1] ✉

Extracellular DNA traps (ETs) represent an immune response by which cells release essential materials like chromatin and granular proteins. Previous studies have demonstrated that the transdifferentiation of vascular smooth muscle cells (VSMCs) plays a crucial role in atherosclerosis. This study seeks to investigate the interaction between CD68+ VSMCs and the formation of ETs and highlight its function in atherosclerosis. Here we show that ETs are inhibited, and atherosclerotic plaque formation is alleviated in male *Myh11^Cre-Pad4^flox/flox* mice undergoing an adeno-associated-virus-8 (AAV8) mediating overexpression of proprotein convertase subtilisin/kexin type 9 mutation (PCSK9) injection and being challenged with a high-fat diet. Obvious ETs generated from CD68+ VSMCs are inhibited by Cl-amidine and DNase I in vitro. By utilizing VSMCs-lineage tracing technology and single-cell RNA sequencing (scRNA-seq), we demonstrate that the ETs from CD68+ VSMCs influence the progress of atherosclerosis by regulating the direction of VSMCs' transdifferentiation through STING-SOCS1 or TLR4 signaling pathway.

Extracellular DNA traps (ETs) are reported as the innate immune response by which cells release DNA to extracellular space, forming the web-like structure composed of depolymerized chromatin, citrullinated histone, and cellular proteins[1]. Typically, ETs are thought to immobilize and kill microorganisms. The formation of ETs was first discovered in neutrophils, named neutrophil extracellular traps (NETs), by Brinkmann et al and believed to be an efficient defense response for eliminating microorganisms[2]. Besides neutrophils, other myeloid cells, including macrophages, mast cells, eosinophils, and basophils have been shown to produce ETs structures in different diseases[3–5], including atherosclerosis[6].

Accumulating studies support that ETs also have important roles in noninfectious and sterile diseases, such as systemic lupus erythematosus, vasculitis, etc.[7]. However, findings on NETs in atherosclerosis are inconsistent. Autopsied plaques from patients with coronary heart disease exhibit prominent NETs[8]. Knight et al found that chemical inhibition of NETs using Cl-amidine, a peptidyl arginine deiminase (PAD) inhibitor, mitigated vascular inflammation and inhibited plaque progression in murine models of atherosclerosis[9]. A recent study confirmed this result in *Apoe^-/-Pad4^-/-* mice[10]. However, they found that Cl-amidine could not inhibit NETs formation induced by cholesterol crystals even if histone citrullination was blocked. Moreover, by transplanting the bone marrow of *Pad4^-/-* mice into *Ldlr^-/-* mice, Franck et al did not find the influence of impaired NETs on plaque burden in a chimera atherosclerotic model[11]. These inconsistent results implied that ETs originating from non-myeloid cells could be responsible for plaque progression, as global inhibition of ETs had consistent protective effects on atherogenesis in atherosclerotic models.

[1]Department of Cardiology, Shanghai Tenth People's Hospital, Tongji University, School of Medicine, Shanghai, China. [2]Biomedical Research Center, Zhongshan Hospital Institute of Clinical Science, Fudan University, Shanghai, China. [3]Department of Endocrinology, Xinhua Hospital, Shanghai Jiaotong University, School of Medicine, Shanghai, China. [4]Cardiovascular Division, Department of Medicine, Brigham and Women's Hospital, Harvard Medical School, Boston, MA, USA. [5]These authors contributed equally: Ming Zhai, Shiyu Gong. ✉e-mail: pwenhui@tongji.edu.cn

It is known that atherosclerotic plaque mainly consists of endothelial cells, vascular smooth muscle cells (VSMCs), and inflammatory cells[12]. The VSMCs in the atheroma can transform into an intermediate cell state or different types of cells, such as harmful macrophage-like cell phenotypes or so-called beneficial "Fibro-myocyte" or "Fibro-chondrocyte" phenotypes[13]. However, the potential mechanisms regulating the transition of VSMCs under specific microenvironments are still unknown. Macrophages are the primary inflammatory cells in atherosclerotic lesions, which can also generate ETs, a process called named macrophage extracellular traps (METs). METs have been reported to accelerate the progression of the disease, such as rhabdomyolysis-induced acute kidney injury or autoimmune arthritis[14,15]. Previous studies using lineage tracing methods confirmed that VSMCs transdifferentiate into "inflammatory" macrophage-like cells expressing CD68 (CD68[+] VSMCs) and serve as the main resource of macrophages in advanced atherosclerotic plaque[16,17]. Therefore, we hypothesized that VSMCs were activated and transformed into CD68[+] VSMCs within the atherosclerotic plaque, produced ETs, and played a key role in plaque progression.

This study aims to investigate whether the existence of ETs stemmed from CD68[+] VSMCs in plaque and their influence on the progression of atherosclerotic lesions. To shorten the period of atherosclerosis and minimize the severe impact of genetic defects or deformities caused by the polygenic variation[18], we overexpressed the PCSK9 by using the AAV-PCSK9 in mice. Also, we ligated left carotid arteries to accelerate the atherosclerotic progress as previously reported[19]. Moreover, using VSMCs-lineage tracing mice and single-cell RNA sequencing (scRNA-seq), we identified the effects and mechanisms of ETs from CD68[+] VSMCs on the plaque.

## Results

### ETs localized mainly with macrophages in advanced plaque
First, we compared the ETs generated by neutrophils (NETs) and macrophages (METs) in mice aortic roots[20]. Interestingly, all METs or NETs releases were closely associated with the evolution of atherosclerosis plaque. The proportion of NETs gradually diminished while METs increased in a time-dependent manner. Notably, in advanced plaques, the ETs were mainly associated with macrophages (90%, $P < 0.01$) instead of neutrophils (Fig. 1a–c). Strikingly, the number of ETs positive macrophages was nearly 5 times ($P < 0.01$) higher than the number of ETs positive neutrophils in advanced plaque (Fig. 1d, e). The ETs staining mainly co-localized with macrophages in the necrotic core of plaque and VSMCs on the fibrous cap (Fig. 1f, g). In addition, the flow cytometry results confirmed that ETs[+] α-SMA[+] CD68[+] cells accounted for the largest percentage (60%, $P < 0.01$) of CD68[+] cells with ETs[+] in advanced plaques. Furthermore, according to the IF and oil red O results, the increased ratio of α-SMA, CD68, and citrullinated histone H3 (H3CIT) triple-positive areas (10% to 70%) was correlated with the progression of atherosclerotic plaque (Supplementary Fig. 1a–d and i–k).

### VSMCs-lineage tracing technology revealed that CD68[+] VSMCs generated ETs
Previous studies reported that VSMCs in plaques expressed CD68, suggesting that the ETs[+] α-SMA[+] CD68[+] cells might be generated from VSMCs[16]. To track ETs[+] CD68[+] VSMCs origin, we developed a VSMCs-lineage tracing murine model $B6-G/R\ Myh11^{Cre}$. VSMCs and their progenies permanently expressed Tdtomato after induction by tamoxifen. Then, Tdtomato[+] VSMCs expressed red fluorescence under a confocal microscope. Here, Tdtomato[+] VSMCs within the plaque expressed PAD4 at a greater level ($P < 0.01$) than Tdtomato[+] VSMCs within the normal aortic medial layer, indicating that VSMCs could potentially produce ETs under atherosclerotic conditions (Supplementary Fig. 2c–f). Using m-IF, we found that the CD68[+] Tdtomato[+] VSMCs generated ETs as assessed by the colocalization of diffuse DNA

and H3CIT in Tdtomato[+] cell-rich areas within lesions. (Figs. 1h, i and Supplementary Fig. 2g).

### Ox-LDL could stimulate ETs generated by CD68[+] VSMCs in vitro
The oil red O staining and CCK8 results indicated the highest ratio of positive cells and viability (80%) at 72 h (Figs. 2a–c). In addition, the cholesterol-loaded rat aortic vascular smooth muscle cells (RASMCs) had higher mRNA expression levels of the macrophage markers such as CD68, CX3CR1, and MAC2, along with a 2-fold reduction of expression of the VSMCs contraction markers ACTA2, MYH11, and CNN1 compared with the control group (Fig. 2d). The protein level changes of α-SMA (50% reduction, $P = 0.01$) and CD68 (20% increase, $P < 0.01$) were also confirmed by WB (Figs. 2e–g). Meanwhile, cholesterol-loading RASMCs had more phagocyted bioparticles and higher expression of phagocyted genes than the control group (Supplementary Fig. 3a, b). Moreover, we used PDGF to induce RASMCs' dedifferentiation. Then we compared expression levels of the PADs family's mRNA among RASMCs, transdifferentiated RASMCs, and dedifferentiated RASMCs induced by PDGF, respectively. PAD4 was mostly elevated (2.6 fold, $P < 0.01$) among PAD family members unless RASMCs transdifferentiated into CD68[+] VSMCs, instead of RASMCs' dedifferentiation. At the same time, stem-cell-related genes' expressions, such as OCT4 were lower in transdifferentiated RASMCs compared with dedifferentiated RASMCs (Supplementary Fig. 3c, d). Consistent with this, PAD4 expression upregulated robustly in RASMCs after cholesterol-loaded assessed by WB ($P < 0.01$) (Fig. 2i, j) and IF (Fig. 2k). Considering that the citrullination of chromatin histones was a necessary step in forming ETs[5], we chose H3CIT as an IF staining marker in vivo and as a quantitative marker for WB in vitro. Considering oxidatively modified low-density lipoprotein (ox-LDL) was extensively studied as a risk factor for atherosclerosis development, we wondered if ox-LDL could also induce ETs. We found that ox-LDL could induce CD68[+] VSMCs to generate ETs in vitro as assessed by WB, dsDNA concentration in the supernatant, and IF staining of the ETs. After ox-LDL stimulation with 100 μg/ml for 8 h, the CD68[+] VSMCs had increased expression of H3CIT protein (3 fold up-regulated, $P < 0.01$) and released dsDNA (1000 ng/ml increased, $P < 0.01$). In addition, we also observed noteworthy fibers, mainly containing DNA and protein of H3CIT, extending from the cell nucleus to the extracellular (Fig. 2l–o and Supplementary Fig. 3e–h). Following the induction of ox-LDL, when ETs[+] CD68[+] VSMCs were treated with DNase I, the generation of ETs was inhibited with decreased H3CIT protein levels (50% reduction), and the DNA release was less than 1000 ng/ml (Fig. 2p–r). To explore if PAD4 expression was essential for ETs, we used Cl-amidine and PAD4-specific siRNA to inhibit the expression of PAD4 before ox-LDL stimulation on CD68[+] VSMCs. We observed that the H3CIT protein level was reduced by more than 30% ($P < 0.01$) (Fig. 2s–u and Supplementary Fig. 3i–k).

### PAD4 deficiency in VSMCs reduced fatty streak formation and instability
Because the ETs release was reduced when PAD4 was inhibited in CD68[+] VSMCs, we explored the effect of inhibition of CD68[+] VSMCs generated ETs' on the progression of atherosclerosis by using VSMCs-specific PAD4 conditional knockout ($Myh11^{Cre}Pad4^{flox/flox}$) mice. The groups of HFD-fed $Myh11^{Cre}Pad4^{flox/flox}$ and $Pad4^{flox/flox}$ mice had similar weights and blood glucose, TC, TG, HDL, and LDL concentrations (Fig. 3a and Supplementary Fig. 4a–d). After being sacrificed at the point of 12 or 24 weeks, we were surprised to find that loss of PAD4 exclusively in VSMCs resulted in a significant 40% reduction of lesion size ($P < 0.01$) at the point of 24 weeks (Fig. 3b–f). The concentration of plasma dsDNA also decreased by nearly 70% in $Myh11^{Cre}Pad4^{flox/flox}$ mice (Fig. 3g). Finally, the IF staining of α-SMA, CD68, and H3CIT revealed that the inhibition of PAD4 in VSMCs suppressed the ETs released from CD68[+] VSMCs ($P < 0.001$) in $Myh11^{Cre}Pad4^{flox/flox}$ without

altering the ETs produced by myeloid macrophages in the brachio-cephalic artery (BCA) lesions or in aortic roots lesions (Figs. 3h–l and Supplementary Fig. 4e–h). Such change indeed reduced the ratio of CD68+ VSMCs of total CD68+ cells (Supplementary Fig. 4i). Further-more, we observed a 10–20% necrotic area reduction (Fig. 3m, n and

Supplementary Fig. 4k), a 2-fold increase in fibrous area (Fig. 3o, p, and Supplementary Fig. 4l), a 2-fold reduction in CD68+ area (Fig. 3q, r and Supplementary Fig. 4m), and a 40% reduction in MMP9+ area (Fig. 3s, t, and Supplementary Fig. 4n) (all *P* < 0.01) in BCA lesions or aortic root lesions from *Myh11*<sup>Cre</sup>*Pad4*<sup>flox/flox</sup> mice compared with that

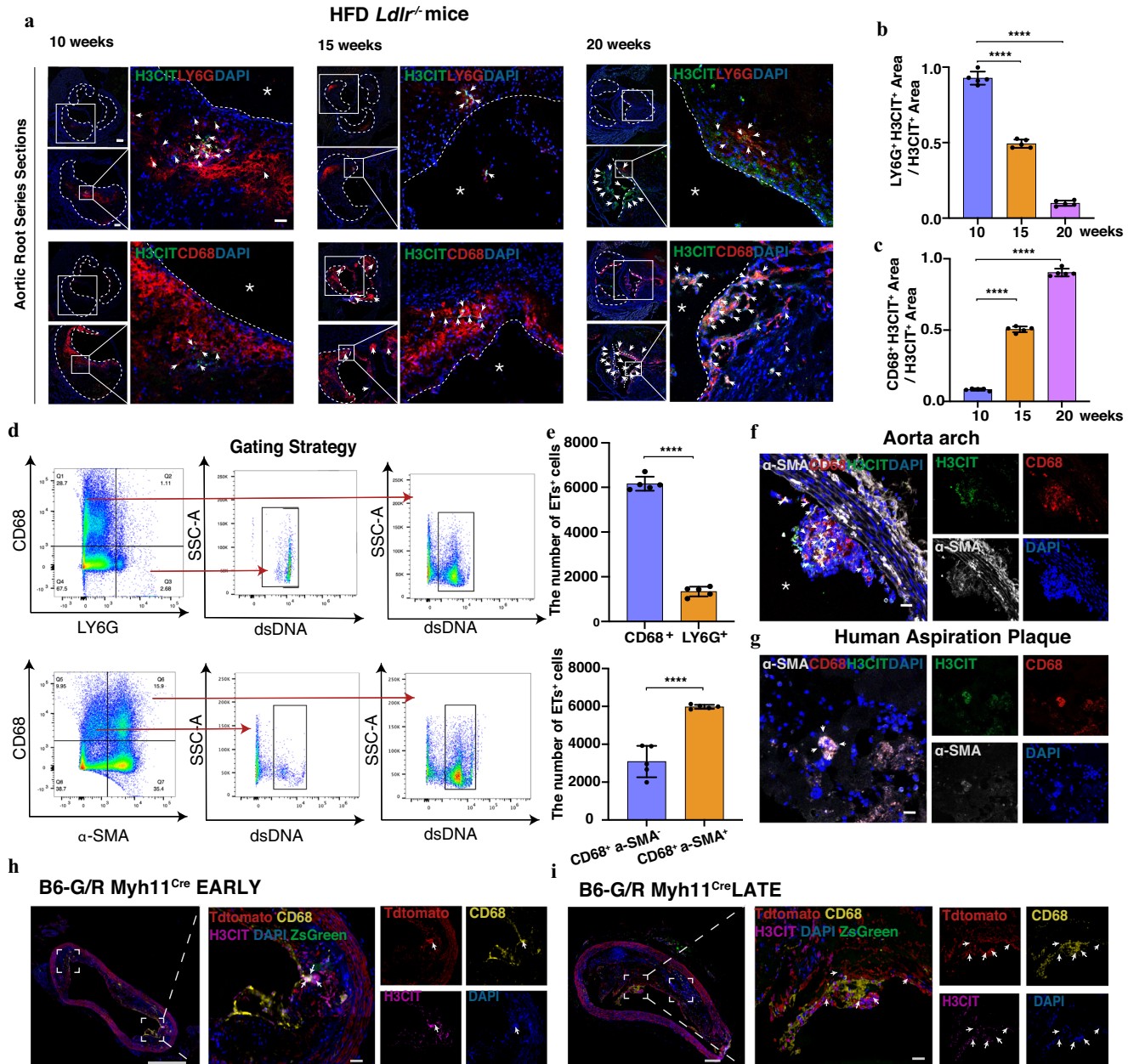

**Fig. 1 | Neutrophil extracellular traps (NETs) or Macrophage extracellular traps (METs) were Generated in a Time-dependent Manner in the Development of Atherosclerosis Plaque. a** Regions of 10 weeks, 15 weeks, and 20 weeks HFD-fed mice's adjacent series sections plaque had neutrophils (LY6G) co-located with H3CIT, or macrophages (CD68) co-localized with citrullinated histone H3 (H3CIT). Scale bar = 200 μm, 100μm, 25μm respectively. The white arrow showed the H3CIT+ cells of Macrophages or Neutrophils. **b** Quantification of the ratio of ETs+ neutrophil area/ETs+ total area within plaque lesions harvested from three-time points HFD fed *Ldlr*-/- mice (each time point, *n* = 5 mice). (10w vs 15w ****p < 0.0001; 10w vs 20w ****p < 0.0001) **c** Quantification of ETs+ macrophage area/ETs+ total area (each time point, *n* = 5 mice). (10w vs 15w ****p < 0.0001; 10w vs 20w ****p < 0.0001). **d** Gating strategy for CD68+ ETs+ macrophages, LY6G+ ETs+ Neutrophils, and α-SMA+ CD68+ ETs+ VSMCs. **e** Quantification of CD68+ ETs+ macrophages, LY6G+ ETs+ neutrophils, and α-SMA+ CD68+ ETs+ vascular smooth muscle cells (VSMCs) in

the plaque harvested from *n* = 5, 24 weeks HFD fed mice. (CD68+ ETs+ vs LY6G+ ETs+ ****p < 0.0001; α-SMA+ CD68+ ETs+ vs α-SMA- CD68+ ETs+ ****p < 0.0001). **f** Representative images within plaque from HFD fed *Ldlr*-/- mice's ascending aorta. Scale bar = 30 μm. The white arrow showed the α-SMA+ H3CIT+ CD68+ cells. **g** Immunofluorescence staining (IF) within plaque from human artery aspiration plaque. Scale bar = 10 μm. **h, i** IF staining of H3CIT and CD68 on the early or late stage of aortic arch plaque harvested from *B6-G/R Myh11*<sup>Cre</sup> mice fed on HFD diet, Scale bar = 200 μm, 50 μm, respectively. White arrows are pointed at the H3CIT positive cells within the plaque. The side of the white star represents the lumen side. For all panels, error bars represent SD. *p*-value was determined by unpaired two-tailed Student's *t*-test (**e**) or one-way ANOVA with Bonferroni post-test (**b, c**). Source data are provided as a Source Data file. Each experiment was repeated independently 3 times for (**f–i**).

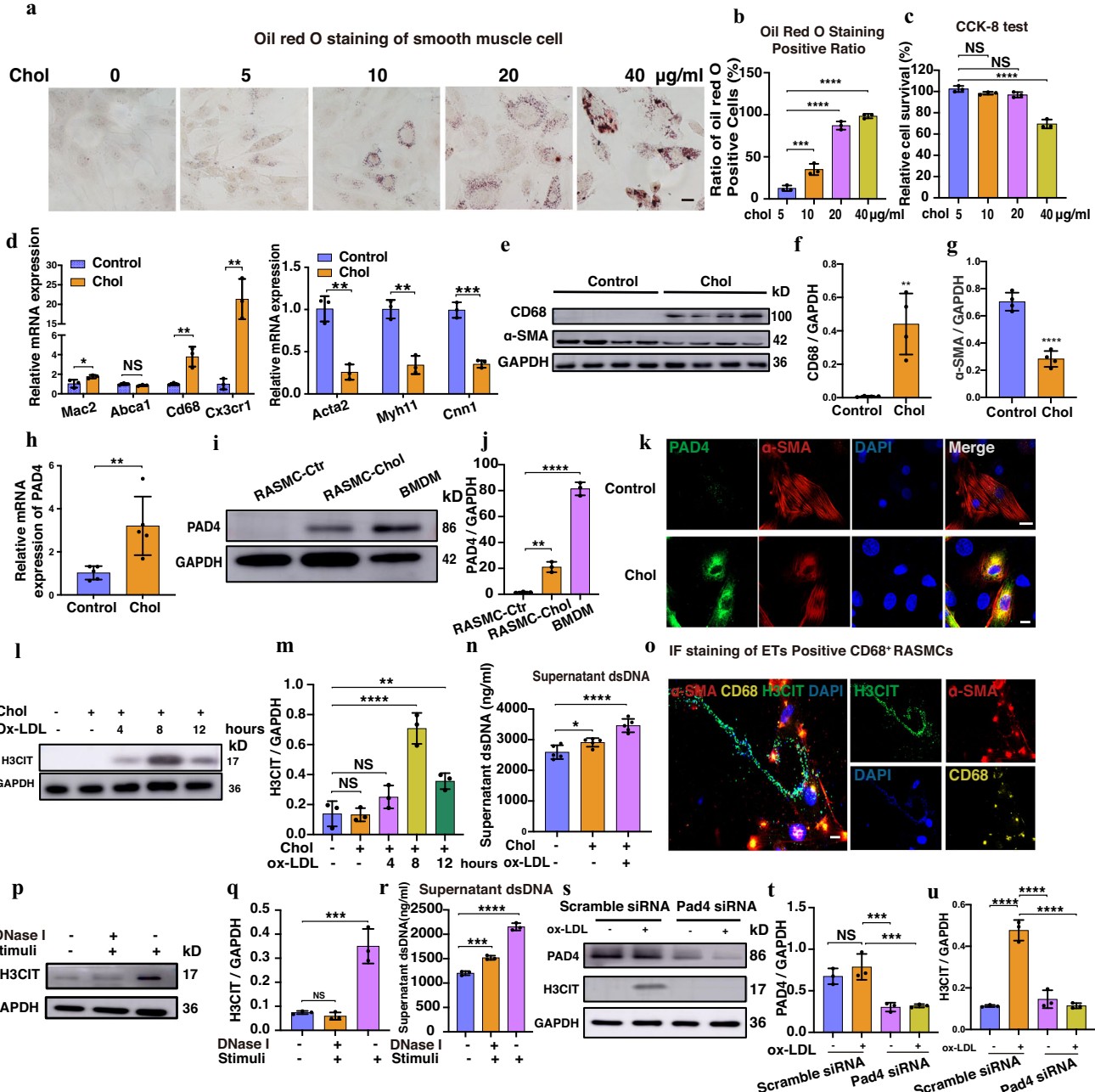

**Fig. 2 | Low-density lipoprotein induced CD68⁺ VSMCs generated ETs in Vitro.**
**a** Oil red O staining in VSMCs. Scale bar = 50 μm. **b** Quantitation of a (*n* = 3 independent experiments). (5 vs 10, ***p = 0.0009, 5 vs 20, ****p < 0.0001; 5 vs 40, ****p < 0.0001). **c** CCK8 results (*n* = 3 independent experiments) (5 vs 10 *p* = 0.2341, 5 vs 20 *p* = 0.0827, 5 vs 40****p < 0.0001). **d** mRNA expression of Cd68 (**p = 0.0089), Abca1 (*p* = 0.1439), Mac2 (*p = 0.0466), and Cx3cr1 (**p = 0.0024), Acta2 (**p = 0.0018), Myh11 (**p = 0.0017), and Cnn1(***p = 0.0004) (*n* = 3 independent experiments). **e** CD68, α-SMA expression levels. **f** (**p = 0.0032) and **g** (****p < 0.0001), quantitation of **e** (*n* = 4 independent experiments). **h** PAD4 mRNA expression (*n* = 5 independent experiments). (**p = 0.0082). **i** the Expression level of PAD4 of different cells. **j** Quantitation of i (*n* = 3 independent experiments). (**p = 0.0013; ****p < 0.0001). **k** IF of PAD4 and α-SMA. Scale bar = 20 μm. **l** H3CIT's expression of different time points. **m** Quantitation of l (*n* = 3 independent experiments). (NS *p* > 0.05, ****p < 0.0001; **p = 0.0097). **n** Released dsDNA concentration. (*n* = 5 independent experiments). (****p < 0.0001; *p = 0.0472). **o** IF of ETs from CD68⁺ VSMCs. Scale bar = 20 μm. **p** The ETs were suppressed by DNase I. **q** Quantitation of p (*n* = 3 independent experiments).NS *p* > 0.05, ***p = 0.0004. **r** Supernatant dsDNA concentration after DNase I intervened (*n* = 3 independent experiments). (****p < 0.0001 and ***p = 0.0008). **s** After siPAD4 intervened, ETs were suppressed. **t**, **u** Quantitation of s (*n* = 3 independent experiments) **t**, NS *p* > 0.05, ***p = 0.0007 and ***p = 0.0008; **u**, ****p < 0.0001. For all panels, error bars represent SD. Source data are provided as a Source Data file. *p*-value was determined by unpaired two-tailed Student's *t*-test (**d**, **f**–**h**) or one-way ANOVA with Bonferroni post test (**b**, **c**, **j**, **m**, **n**, **q**, **r**, **t**, **u**). Each experiment was repeated independently three times for (**k**, **o**). Cholesterol (chol).

from the control group of mice. Furthermore, we also find evidence that ETs generated from CD68⁺ VSMCs in three specimens from human aspiration atherosclerosis plaque and find that areas of ETs positive CD68⁺ VSMCs are related to MMP9 positive areas in a series adjacent section (Supplementary Fig. 4o). These findings highlighted that CD68⁺ VSMCs were essential for ETs production in advanced plaques, which played a crucial role in accelerating plaques' instability and atherosclerosis progression.

## Specific PAD4 deficiency of VSMCs reduced ETs generation and influenced VSMCs' transdifferentiation

Considering the percentage of CD68[+] VSMCs were lower in *Myh11Cre·Pad4flox/flox* mice compared with *Pad4flox/flox* mice. We hypothesized that the decrease in the total number of macrophages could be associated with reducing CD68[+] VSMCs after ETs' reduction. VSMCs lineage tracing mouse *B6-G/R Myh11CrePad4flox/flox* was generated, and *B6-G/R Myh11Cre* mice served as controls (Fig. 4a). Similar results were demonstrated for suppressed plaque size and lipids content in *B6-G/R Myh11CrePad4flox/flox* mice (Fig. 4b, c Supplementary Fig. 5a–c). In addition to these, the dsDNA concentration in plasma was also lower (1500 ng/ml reduction, $P < 0.01$) in the *B6-G/R Myh11CrePad4flox/flox* group compared with the control group (Fig. 4d). Histologically, knocking down PAD4 in VSMCs abolished the ETs generated from Tdtomato[+] CD68[+] cells and altered the cells' transdifferentiation to CD68[+] cells (Fig. 4e–g and Supplementary Fig. 5d–k). Yet, in vitro knocking down PAD4 within dedifferentiated RASMCs did not influence cell behaviors such as apoptosis, migration, proliferation, and lipid overload (Supplementary Fig. 5l–r). Thus, ETs were released by CD68[+] VSMCs, but not the dedifferentiated VSMCs, and they played a critical role in regulating VSMCs' transdifferentiation into CD68[+] VSMCs, thereby influencing plaque burden and stability.

## ScRNA-seq combined with VSMCs-lineage tracing technology revealed cell mapping changes after inhibition of ETs' releases

To gain insights into the effects of PAD4 deficiency on the VSMCs' transdifferentiation, we performed scRNA-seq of pooled Tdtomato[+] cells and ZsGreen[+] cells. The 2 groups of cells were sorted from the aortas of HFD-fed *B6-G/R Myh11CrePad4flox/flox* mice and *B6-G/R Myh11Cre* mice (each group was 6 mice). Following quality control and doublet exclusion, unsupervised clustering and uniform manifold approximation and projection (UMAP) projection of 35,511 cells (Tdtomato[+] cells: 16,130; ZsGreen[+] cells: 19,381)[21] resulted in 10 clusters representing different cell subsets that were annotated based on differentially expressed genes (Supplementary Fig. 6)[16]. Our scRNA-seq results revealed dramatic changes in the Tdtomato[+] cells expression landscape upon VSMCs PAD4 deficiency, while the evolution of ZsGreen[+] cell mapping was minimal, as shown in Fig. 4h–j.

## ETs generated from CD68[+] VSMCs influenced the direction of VSMCs' trans-differentiation from beneficial-like cells to harmful-like cells

Based on the above scRNA-seq data, we performed the sub-clustering of Tdtomato[+] cells to dissect further changes in the context of PAD4 deficiency in VSMCs. Sub-clustering revealed 10 clusters identified based on the expression of defining markers (Figs. 4k, l and Supplementary Fig. 7a, b). The senescent VSMCs cluster showed high expression of Vcam1 and Ccl2, with GO enrichment analysis results of "SMC apoptotic progress" (Supplementary Fig. 7c), resembling those previously described as senescent VSMCs[22]. The ratio of senescent VSMCs dramatically reduced following PAD4 knockout in VSMCs (Fig. 4m), consistent with the results of IF staining of apoptotic RASMCs (Supplementary Fig. 7d, e). The cluster of stem-like SMC resembled "SEM cells" and was further characterized by elevated expression of Klf4 and Vcam1[14]. The Stem-Like SMCs marker genes were enriched in "Smooth Muscle Cell Differentiation". Marker genes of clusters of Fibro-myocyte and Fibro-chondrocyte (beneficial cells) were enriched in "Muscle Structure Development," participating in the process of plaque stability. Moreover, marker genes of clusters of PAD4 (hi) CD68[+] SMC, CD68[+] SMC, and senescent SMC (harmful cells) were enriched in "Inflammation response to wounding" (Shown in Fig. 5a), generating inflammation and instability. The ratio of "Stem-like SMC" and "beneficial cells" upregulated while the ratio of "harmful cells" downregulated in VSMCs knocked out PAD4 within atherosclerosis plaque (Fig. 4m). Furthermore, we verified the

bioinformatic analysis results of cells' ratio change by using the mFISH probe (Klf4, Ccl2, Pad4, Col3a1, Vim, Acta2, and Spp1) on lineage tracing mice aorta plaque section (Supplementary Fig. 8). As for these, mFISH probe of *Klf4* was used to identify Stem-Like SMC, and Ccl2, PAD4, Col3a1, Vim, Acta2, and Spp1 were used to distinguish senescent SMC, PAD4 (hi) CD68[+] SMC, fibro-chondrocyte, fibroblast, SMCs and transdifferentiated harmful-like cells, respectively. Moreover, the Hist1h1b expression, consistent with cells exclusively released histones[23], was downregulated after PAD4 knockout within Tdtomato[+] VSMCs in advanced plaques (Supplementary Fig. 9a). We performed cell trajectory analysis from Stem-Like SMC to harmful or beneficial cell types and found that PAD4 mainly played roles in CD68[+] SMCs, especially in PAD4 (hi) cell types (Supplementary Fig. 9b–e), instead of dedifferentiated Stem-Like SMCs or intermediate SMCs. Meanwhile, the percentage of PAD4[+] Tdtomato[+] cells was not significantly up-regulated ($P = 0.43$) in the ligated carotid artery (de-differentiated VSMCs) compared with sham (Supplementary Fig. 9f, g). We thus concluded that inhibition of ETs from CD68-positive VSMCs promoted VSMCs' transdifferentiation to beneficial cells instead of harmful cells in the development of plaques within *B6-G/R Myh11CrePad4flox/flox* mice.

## PAD4 (hi) CD68[+] VSMCs could influence surrounding VSMCs by releasing ETs and activating STING-SOCS1 or TLR4 signaling pathway

We screened differently expressed genes (DEGs) of Tdtomato[+] cells harvested from *B6-G/R Myh11CrePad4flox/flox* mice (PadΔ/Δ Tdtomato[+] cells) *vs*. Tdtomato[+] cells harvested from *B6-G/R Myh11Cre* mice (Pad4+/+ Tdtomato[+] cells) (Fig. 5b). Gene Set Enrichment Analysis (GSEA) suggested that "Toll-Like Receptor Signaling Pathway," "STING Signaling Pathway," and "TNF Signaling Pathway" were enriched in *B6-G/R Myh11Cre* Tdtomato[+] cells. Next, we used KnockTF to predict transcription factors that regulated the DEGs. STAT3 ranked first as it regulated 363 upregulated DEGs. Furthermore, violin plots showed several key markers genes within the above signaling pathways between the two groups of Tdtomato[+] cells, such as Gsdmd, Mmp9, and Tnf associated with Toll-like Receptor signaling pathway; and Tbk1, Tmem173 associated with the STING signaling pathway (Supplementary Fig. 10a–c). We then performed bulk RNA-sequencing analysis of RASMCs from the H3CIT + dsDNA challenged group and BSA-treated control group. Different gene expression patterns of cells between 2 groups of cells were shown in (Fig. 5c, d, and Supplementary Fig. 10d, e). We verified bioinformatic analysis results in vitro by treating RASMCs with H3CIT and dsDNA at different time points of ox-LDL stimuli. We found that the phosphorylation of STAT3 was inhibited (by 60%, $P < 0.01$) in H3CIT, and dsDNA challenged the group with STING upregulated compared with the control groups (Fig. 5e–g). Next, we explored STING signaling pathway activation and STAT3 signaling pathway inhibition. We integrated genes regulated by STAT3 (predicted from the STRING website) with DEGs from bulk RNA-seq analysis, as shown in the Venn diagram of Supplementary Fig. 10f. The Socs1 ranked first with the highest log$_2$ fold change value (fold change = 2.95, $P < 0.01$), suggesting it is the key regulator to inhibit STAT3's phosphorylation after STING signaling pathway activation. We found that challenged cells indeed upregulated expression of SOCS1 (increased 50%, $P < 0.01$) (Fig. 5e–g). IF staining showed that the proportion of Ki67[+] Tdtomato[+] SMCs was reduced by 60% ($P < 0.01$) in plaques of B6-G/R *Myh11CrePad4flox/flox* mice compared with the control group (Supplementary Fig. 10g–i). In addition, IF staining of SOCS1 expression in vivo decreased dramatically after inhibition of ETs' releases (Fig. 5l, m, Supplementary Fig. 10j, l). TLR4-MYD88 signaling pathway was also activated in challenged RASMCs (Fig. 5h–j), and the TLR4 regulated downstream protein GSDMD was decreased after inhibition of ETs releases in the plaque of *B6-G/R Myh11CrePad4flox/flox* mice (Fig. 5k, n, o, Supplementary Fig. 10k, m). Furthermore,

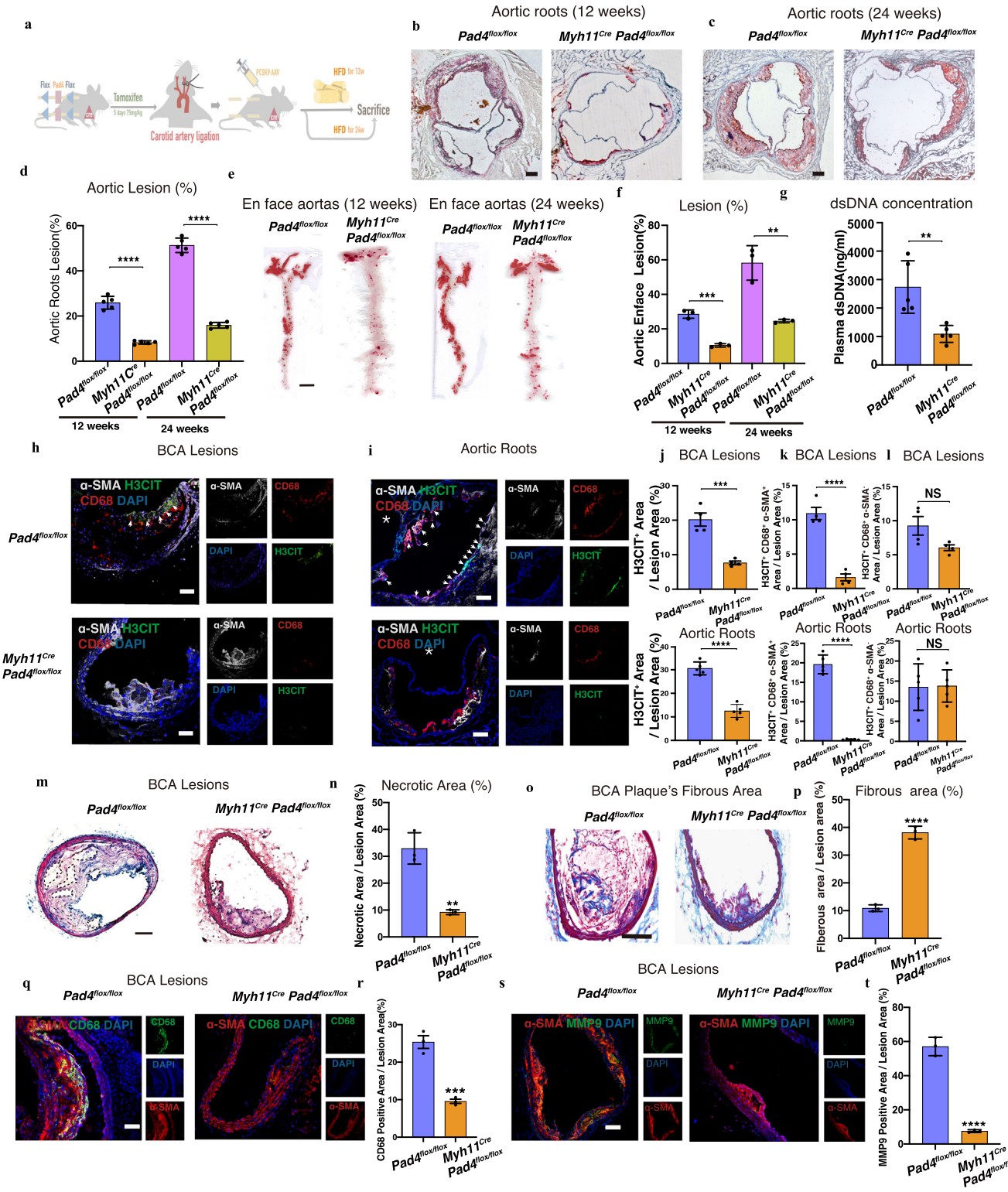

TLR4-induced proliferation of SMCs was actually activated in VSMCs of *B6-G/R Myh11^Cre* mice, consistent with our scRNA-seq results.

## Discussion

In this study, we first discovered that CD68⁺ VSMCs were activated under atherosclerotic risk factors and produced ETs contributed to plaque progression by influencing the microenvironment of advanced plaques.

Megens et al. first reported the presence of NETs in mouse and human atherosclerotic lesions[24]. Since then, several studies revealed

that NETs were not only involved in atherosclerosis[25–27] but also enhanced coagulation and promoted thrombosis[25]. We confirmed the results of previous studies that NETs emerged in the early stage of murine atherosclerotic lesions[28]. However, our experiments further demonstrated that METs prevailed more in advanced plaques, a time frame when the percentage of NETs decreased. In fact, macrophages dominated the immune cell landscape, while neutrophils appeared rare in advanced atherosclerotic lesions[12]. As such, CD68⁺ VSMCs, in addition to regular myeloid-derived macrophages, were the primary inflammatory cells present in the lesions. These results were consistent

**Fig. 3 | VSMCs Specific PAD4 Deficiency Alleviated Atherosclerosis Plaque Formation and Improved the Plaque Stability. a** The procedure of mouse model. **b**, **c** Oil red O-stained results from 2 groups of mice. Scale bar = 100 μm. **d** Quantification b&c (each group and each time point, $n = 5$ mice). ****$P < 0.0001$. **e**, **f** Oil red O-stained aortas of mice (each group and each time point, $n = 3$ mice), and quantification (**f**) ***$p = 0.0003$ **$p = 0.0044$. Scale bar = 0.5 cm. **g** DsDNA concentration within plasma (each group $n = 5$ mice). (**$p = 0.0051$). **h**, **i** IF results in plaque. Scale bar = 20 μm. **j** Ratio H3CIT positive area/lesion area in the brachio-cephalic artery (BCA) (each group $n = 4$ mice) or aortic root (each group $n = 5$ mice). ***$p = 0.0003$. ****$p < 0.0001$. **k** The ratio of H3CIT[+], CD68[+], and α-SMA[+] area/lesion area in the BCA (each group $n = 4$ mice) or aortic root (each group $n = 5$ mice). ****$p < 0.0001$. **l** The ratio of H3CIT[+], CD68[+] and α-SMA[-] area/lesion area in the BCA

(each group $n = 4$ mice), $p = 0.4586$ or aortic root (each group $n = 5$ mice), $p = 0.8971$. **m**, **n** HE staining and ratio of necrotic core area/plaque area in the BCA (each group $n = 3$ mice). Scale bar = 100 μm. The dotted area represents the necrotic area. **$p = 0.0022$. **o**, **p** Masson staining, and the ratio fibrous area/plaque area in BCA (each group $n = 3$ mice). Scale bar = 100 μm. ****$p < 0.0001$. **q**, **r** CD68 stained, and the ratio CD68[+] area/plaque area in BCA (each group $n = 3$ mice). Scale bar = 50 μm. ***$p = 0.0009$. **s**, **t** MMP9 stained, and the ratio MMP9[+] area/plaque area in BCA (each group $n = 3$ mice). ****$p < 0.0001$. Scale bar = 50 μm. NS. means no significance. White arrow means H3CIT positive cells. White star represented the lumen side. All panels, error bars represent SD. Source data are provided as a Source Data file. $p$-value was determined by unpaired two-tailed Student's $t$-test.

with Pertiwi, K. R et al, which found that METs were more numerous than NETs in intact plaques with lipid cores[6]. Moreover, our VSMCs lineage tracing results confirmed that the ETs derived from CD68[+] cells played a dominant role in advanced plaques.

Only a few reported studies have evaluated the roles of ETs generated by CD68[+] cells in chronic disease. Although previous studies demonstrated that treatment by PMA or bacteria on mouse macrophages did not induce ETs[29], we observed that CD68[+] VSMCs generated ETs when stimulated by ox-LDL, suggesting that CD68[+] VSMCs also formed ETs in vitro. In neutrophils and macrophages, elastase and MPO both contribute to the formation of ETs[30,31]. The features of ETs morphology are typically related to their function. For example, the ETs from myeloid cells were illustrated as web-like fibers with MPO and elastase and aligned with the ability to "trap" and kill microorganisms[32]. However, the ETs stretched from CD68[+] VSMCs to neighboring SMCs, suggesting they tended to cause chronic injury within the plaque. Furthermore, we found that ETs from CD68[+] VSMCs were only stained with H3CIT, without MPO or NE, but NETs could be stained with all three classic ETs markers (Supplementary Fig. 2H, I).

PAD4 is expressed mainly in hematopoietic cells, such as neutrophils[33,34]. Usually, PAD4-dependent citrullination of histones induces de-condensation of DNA[35]. Our scRNA-seq and in vitro experiments revealed that the expression of PAD4 changed insignificantly when VSMCs dedifferentiated into intermediate SMCs unless intermediate VSMCs activated into CD68[+] VSMCs. However, other genes in the PAD family changed little after VSMCs' activation. After PAD4 was knocked out specifically in VSMCs, we found a strong reduction of ETs within advanced plaques, alongside decreased pro-inflammatory CD68[+] VSMCs. These results suggested that ETs from CD68[+] VSMCs depended on PAD4 and played a key role in VSMCs activating pro-inflammatory status. Utilizing scRNA-seq to find the specific changes of SMCs fate after inhibition of releases of ETs, our results revealed that the inhibition of ETs decreased the proportion of harmful cells and increased that of beneficial cells, highlighting that ETs from CD68[+] VSMCs mediated the balance of the Janus of VSMCs' transdifferentiation.

Our results indicated that the ETs generated from CD68[+] VSMCs would activate the STING-SOCS1 singling pathway within VSMCs and thus inhibit the phosphorylation of STAT3, which was consistent with DNA-induced damage, as previously reported[36]. STAT3 promoted VSMCs contractile gene expression, proliferation, and inhibition of VSMCs apoptosis[37,38]. Furthermore, STAT3 has been reported to take part in collagen-producing and collagen fibers forming through its phosphorylation, thus playing a key role in plaque stabilization[39,40]. In addition, the TLR4-MYD88 signaling pathway was also activated by ETs and induced a pro-inflammatory[10] or senescent phenotypic change, such as increasing expression of the MMP9, Gsdmd, or Tnf, and induced collagen degradation and plaque instability[41]. Furthermore, CD68[+] VSMCs' proliferation caused by TLR4 activation[42] could be observed in our scRNA-seq results and IF staining in plaque, indicating that the PAD4 (hi) CD68[+] SMC had the potential to maintain its population and mediate positive feedback of pro-

inflammation, thereby promoting atherosclerosis plaque burden (Shown in Fig. 6).

The intermediate cells were once thought to serve as therapeutic targets for atherosclerosis[43]. But the previous study combined with our results confirmed that VSMCs-derived intermediate cells were multipotent and could differentiate into either harmful or beneficial cells[44]. These findings implied that VSMCs-derived intermediate cells were beneficial for lesion stability, and the inhibition of VSMCs' de-differentiation to intermediate cells might be improper. Our study showed that PAD4 (hi) CD68[+] VSMCs could impact surrounding VSMCs by releasing ETs and activating the STING-SOCS1 or TLR4 signaling pathway. ETs regulated the direction of VSMCs' trans-differentiation from beneficial-like cells to harmful-like cells and mediated positive feedback of pro-inflammation. These results suggest that it is more valuable to block the positive feedback of ETs and establish that the CD68[+] VSMCs are served as a more appropriate therapeutic target for atherosclerosis. In conclusion, our study demonstrates that ETs generated from CD68[+] VSMCs adversely contribute to plaque progression and highlight their unexpected role in plaque stability. These findings also provide insights into the trans-differentiation of VSMCs into different intermediate harmful and beneficial subtypes that could influence the composition of plaques and serve as novel therapeutic targets in plaque prevention.

## Methods

### Human atherosclerosis plaque collection

The Ethical Committee of Shanghai Tenth People's Hospital approved our study, and clinical experiments were conducted in compliance with all relevant ethical regulations (Shanghai, China; permit number: 22KN22). Four patients with acute coronary syndrome (ACS) were enrolled, and coronary artery plaque of the patients was obtained from an aspiration device we employed (EXPORT, Medronic lnc). The EXPORT is a quick exchange catheter, a vacuum syringe is connected to the proximal end when the distal tip is advanced on the guidewire until the plaque in the diseased coronary artery. When the vacuum syringe is opened, simple mechanical aspiration removes the plaque from the coronary artery. After we got the plaque, The plaque was fixed with 4% paraformaldehyde for 10 min. Paraformaldehyde-fixed plaque samples were embedded in paraffin blocks and cut into 5 μm thick sections for further analysis. The baseline characteristics of the patients for ACS are listed in Supplementary Table 2. The written informed consent was collected from patients or their relatives.

### Mice research

All animal procedures were approved by the Animal Care and Use Committees of Shanghai Tenth People's Hospital affiliated with Tongji University for animal welfare. Experiments were conducted according to the Guide for the Care and Use of Laboratory Animals published by the National Institutes of Health (NIH Publication, 8th Edition, 2011). The mice were kept in an room with a temperature of 18–25°, relative humidity of 40–70%, and noise below 85 decibels. The room provided an appropriate period of alternating light and dark, usually 12/12 h, to

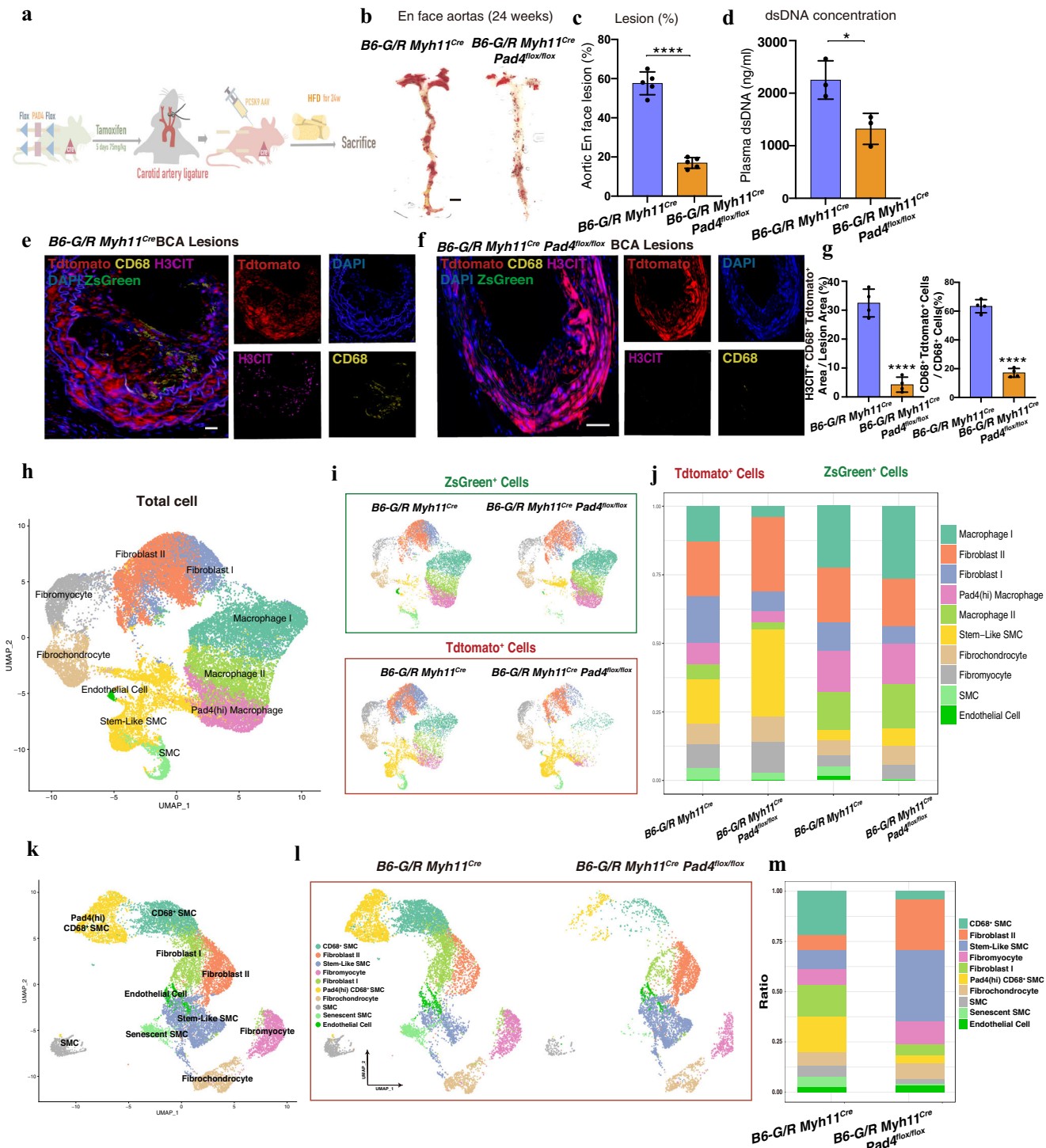

**Fig. 4 | ET Inhibition Reduced the CD68+ VSMCs and scRNA-seq Identified Multiple VSMCs-derived Cell Types During Advanced Atherosclerosis Plaque.** **a** Schematic diagram of the procedure of HFD fed *B6-G/R Myh11^Cre^Pad4^flox/flox^* mice and *B6-G/R Myh11^Cre^* mice for both time points. **b, c** Oil red O-stained aortas isolated from 24 weeks HFD-fed mice between 2 groups and related quantification (each group *n* = 5 mice). ****p < 0.0001. Scale bar = 0.5 cm. **d** The concentration of plasma dsDNA harvested from mice between groups (each group *n* = 3 mice). *p = 0.0265. **e, f** IF staining of H3CIT or CD68 in BCA plaque lesions between groups, respectively. Scale bar = 50 μm. **g** The percentage of H3CIT+ Tdtomato+ and CD68+ area in the total lesion area of the BCAs and the percentage of the number of CD68+ Tdtomato+ cells in the total number of CD68+ Cells (each group *n* = 4 mice). ****p < 0.0001. **h, i** Uniform manifold approximation and projection(UMAP)

visualization of all single cell RNA sequencing (scRNA-seq) data from *B6-G/R Myh11^Cre^* and *B6-G/R Myh11^Cre^Pad4^flox/flox^* mice, including both Tdtomato+ and ZsGreen+ cells. For combined data of 4 samples, representative cell types for each cluster (**h**) and Tdtomato+ or ZsGreen+ status of cell clusters (**i**) were indicated. **j** A diagram of ratio change in Tdtomato+ cells or ZsGreen+ cells between two groups. **k, l** UMAP visualization of all scRNA-seq data of Tdtomato+ cells from *B6-G/R Myh11^Cre^* and *B6-G/R Myh11^Cre^Pad4^flox/flox^* mice. For combined data of 2 samples, representative cell types for each cluster (**k**) and cell clusters from 2 groups (**l**) are indicated. **m** Showed a diagram of ratio changes between two groups within Tdtomato+ cells. For all panels, error bars represent SD. Source data are provided as a Source Data file. *p*-value was determined by unpaired two-tailed Student's *t*-test. BCA: brachiocephalic artery.

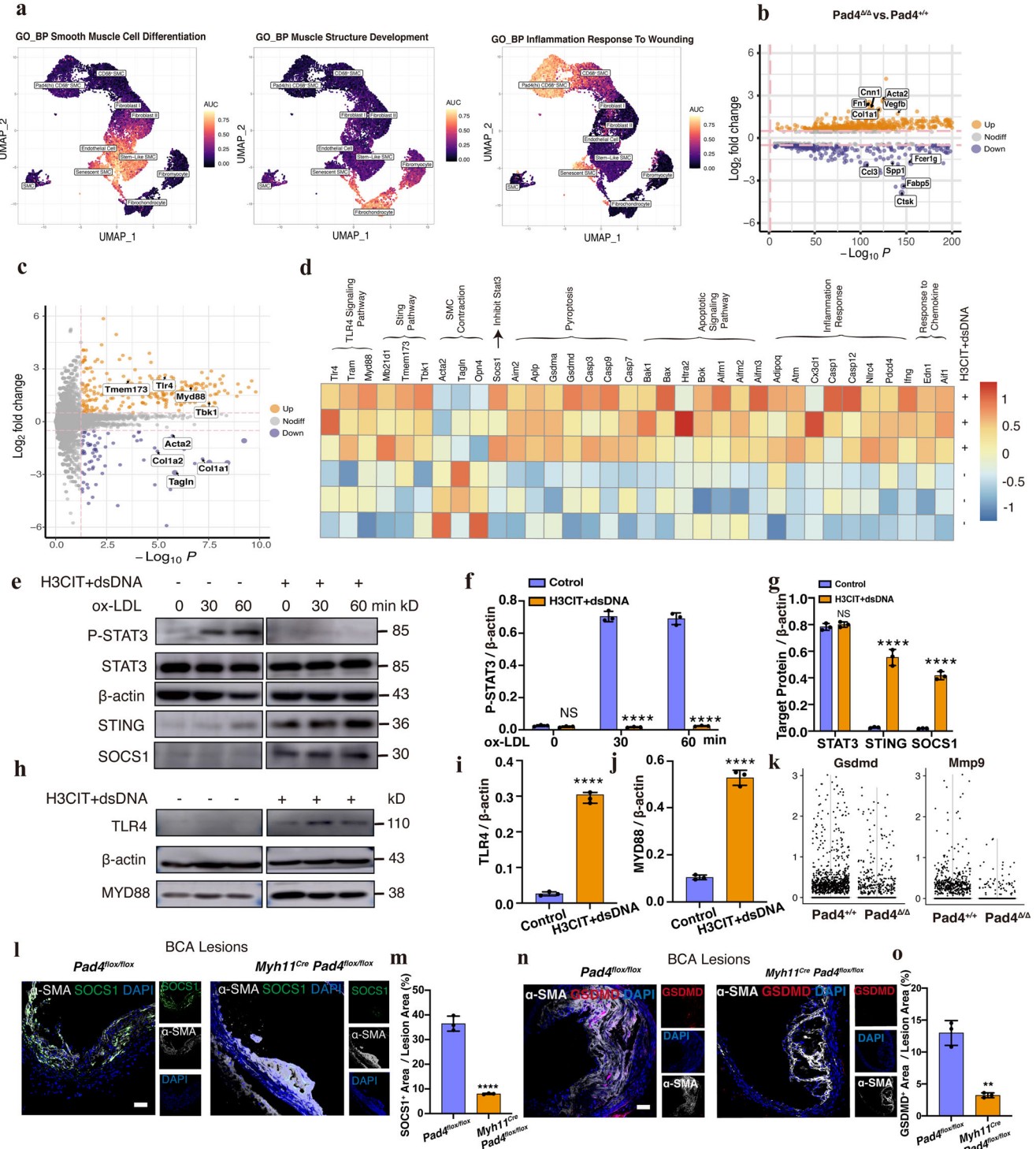

**Fig. 5 | PAD4 (hi) CD68⁺ VSMCs exhibited complex molecular and cellular functions. a** Individual cell area-under-the-curve (AUC) values overlay for selected differential canonical pathway activities. **b** Volcano plots of differentially expressed genes (DEGs) were screened by comparing Tdtomato⁺ cells harvested from *B6-G/R Myh11^Cre Pad4^flox/flox* mice (*Pad4^Δ/Δ* Tdtomato⁺ cells) with that harvested from *B6-G/R Myh11^Cre* mice (*Pad4^+/+* Tdtomato⁺ cells) in scRNAseq data. **c** Volcano plot of DEGs screened by comparing H3CIT + dsDNA challenged rat aortic vascular smooth muscle cells (RASMCs) with control RASMCs of RNA-seq data. **d** Heatmap showed different gene expression patterns between groups of RASMCs in **c**. **e** Western blot analysis showed RASMCs treated with H3CIT + dsDNA induced a marked decrease in the levels of phosphorylated STAT3, and STING-SOCS1 signaling pathway was activated compared with control RASMCs. **f, g** The Quantification of WB results in **e** (*n* = 3 independent experiments). Ox-LDL 0 h *p* = 0.2067, STAT3 *p* = 0.4199, ****p* < 0.0001. **h** Western blot analysis showed RASMCs treated with H3CIT +

dsDNA induced a marked increase in the protein levels of TLR4 and MYD88. **i, j** The Quantification of WB results in **h** (*n* = 3 independent experiments). ****p* < 0.0001. **k** Violin plot of Gsdmd or Mmp9 between two groups of scRNA-seq results. **l** IF staining of SOCS1 in atherosclerosis plaque of BCA lesions of *Pad4^flox/flox* mice and *Myh11^Cre Pad4^flox/flox* mice, respectively. Scale bar = 50 μm. **m** The Quantification of the ratio of SOCS1 positive area in plaque lesion area between 2 groups (each group *n* = 3 mice). ****p* < 0.0001. **n** IF staining of GSDMD in atherosclerosis plaque of BCAs of *n* = 3 *Pad4^flox/flox* mice and *n* = 3 *Myh11^Cre Pad4^flox/flox* mice, respectively. Scale bar = 50 μm. **o** The Quantification of the ratio of GSDMD positive area in plaque lesion area between 2 groups (each group *n* = 3 mice). *p* = 0.001.NS. means no significance. The side of the white star represents the lumen side. For all panels, error bars represent SD. Source data are provided as a Source Data file. *p*-value was determined by unpaired two-tailed Student's *t*-test.

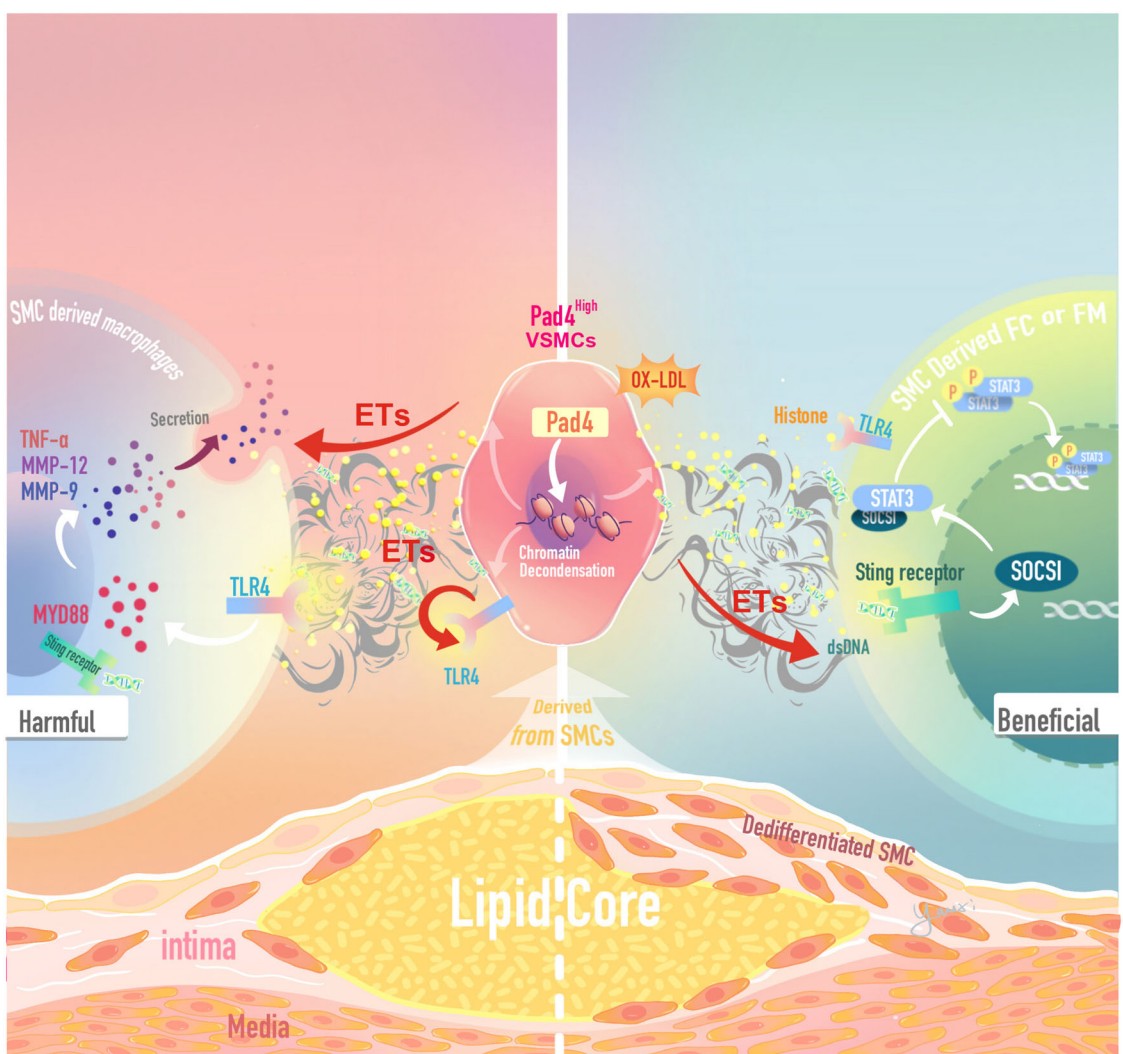

**Fig. 6 | Mechanism diagram: PAD4 (hi) CD68⁺ VSMCs could activate surrounding VSMCs by releasing ETs.** PAD4 (hi) CD68⁺ VSMCs released ETs preferentially contributed to trans-differentiation of VSMCs to CD68⁺ VSMCs and increased collagen degradation, rather than VSMCs' trans-differentiation into Fibrochondrocyte (FC) or Fibromyocyte (FM) to stabilize plaques. The ETs activated the STING-SOCS1 signaling pathway, and TLR4-MYD88 signaling pathway of surrounding VSMCs derived cells within the microenvironment of atherosclerosis plaque. The TLR4 signaling pathway activation resulted in the releases of TNF-α, MMP-9, and MMP-12, thus, causing collagen degradation. STING-SOCS1 signaling pathway activation inhibited the process of phosphorating STAT3, thus contributing to the inhibition of VSMC's contractile phenotype and transferring to harmful-like cells, resulting in plaque progression or instability.

simulate the normal circadian rhythm of mice. Male *Ldlr* $^{-/-}$, *Myh11-CreER*$^{T2}$ (*Myh11*$^{Cre}$)[45], *Pad4*$^{flox/flox}$ (purchased from Cyagen Biosciences), *CAG-LoxP-ZsGreen-Stop-LoxP-Tdtomato (B6-G/R)* mice (purchased from Shanghai Model Organisms), were used in our present study. *Ldlr*$^{-/-}$, *Myh11-CreER*$^{T2}$, *Pad4*$^{flox/flox}$, and *B6-G/R* mice were genotyped by PCR based on the protocol provided by the company. For *Myh11-CreER*$^{T2}$ mouse models, Cre recombinase was activated in male mice with 75 mg/kg tamoxifen (T-5648, Sigma) intraperitoneal injections for 5 days at the age of 8 weeks. Male littermate controls were used for all studies. The above mice were administered with a tail-vein injection of recombinant adeno-associated virus (AAV, containing $1.0 \times 10^{11}$ genome copies) encoding PCSK9 (AAV8.ApoEHCR-hAAT.D377Y-mPCK9.bGH) to induce atherosclerosis[19,46]. For the mice's carotid ligation: mice were anesthetized with the intraperitoneal injection of ketamine (80 mg/kg) and xylazine (5 mg/kg). The left carotid artery was completely ligated with a 6−0 silk suture proximal to the carotid bifurcation. Mice were fed with Clinton/Cybulsky High Fat Rodent Diet (HFD, 40% calories from fat; and contains 1.25% cholesterol, D12108C, Research Diets). Body weight and food intake were measured every week. Mice were randomly divided into experimental groups for the various measurements based on genotyping. Mice were sacrificed by cervical dislocation and then perfused via the left ventricle: 5 ml phosphate-buffered saline (PBS), 10 ml 4% paraformaldehyde, and 5 ml PBS. Ascending arteries and hearts were carefully dissected and frozen-embedded in the Optimal Cutting Temperature compound before sectioning and further analysis. A total number of 25 male 8 weeks aged *Ldlr*$^{-/-}$ mice, a total number of 30 male 8 weeks aged *Pad4*$^{flox/flox}$ *Myh11*$^{Cre}$ mice, a total number of 30 male 8 weeks aged *Pad4*$^{flox/flox}$ mic, and a total number of 30 male 8 weeks aged *B6G/R Myh11*$^{Cre}$ mice, and a total number of 30 male 8 weeks aged *B6-G/R Myh11*$^{Cre}$*Pad4*$^{flox/flox}$ mice participated in our research. As for the animals' euthanasia practices, we anesthetized the mice with the intraperitoneal injection of ketamine (80 mg/kg) and xylazine, then the animals were sacrificed by cervical dislocation.

**Atherosclerotic lesion analysis**
For oil red O staining of en-face aortas, the aorta from the aortic arch to the iliac artery was carefully isolated integrally. The vessels were cut

and flattened along the longitudinal axis. Aortas were fixed in 4% formalin for 48 h, then washed in PBS, incubated in oil red O solution for 1 h at room temperature, and washed in 85% propylene glycol and PBS three times on a shaking table. After pinning the aortas between the cover glass and slide glass with glycerin gelatin, an electronic scanner was used to scan and obtain images. Furthermore, the aortic root and BCAs were cut into sections and incubated in oil red O solution for 1 h. The necrotic core contents were quantified with HE staining, and the collagen contents were quantified with Masson staining. All analyses were assessed by Image J software (V2.10, NIH, USA)

## Plasma index measurement

Blood samples were collected in heparinized tubes, incubated on ice for 10 min, and centrifuged at 300 g for 10 min at 4 °C. The obtained plasma samples were subjected to determination for the concentration of glucose, Total Triglyceride (TG), Total Cholesterol (TC), High-Density Lipoprotein-Cholesterol (HDL), and Low-Density Lipoprotein-Cholesterol (LDL). A Hitachi blood autoanalyzer performed all biochemical analyses.

## Rat research for ex vivo research

Previous studies have reported that primary aortic vascular smooth muscle cells isolated from rats were more accessible and viable than those isolated from mouse aorta[47]. Thus, rat aortic vascular smooth muscle cells (RASMCs) from 25 male Sprague-Dawley rats (200–250 g) were used for vitro experiments. After being cleaned in PBS solution, the aortas were incubated in Hank's solution with 180 units/mL of collagenase at 37 °C for 30 min, and then the adventitia and endothelium were removed. Finally, aortas were incubated for 12 h in DMEM (SH30243.01, HyClone) with 10% FBS (10099141, Gibco) at 37 °C with 95% $O_2$ and 5% $CO_2$. The purity of RASMCs harvested was confirmed by IF staining RASMCs' marker: α-SMA (ab7817, Abcam, 1:100). As for the isolation of bone marrow cells (BMDMs), after the rat was sacrificed, the BMDMs were isolated from the femurs of adult rat and differentiated into BMDMs using 40 ng/ml M-CSF (HY-P7247, MCE) plus 10% FBS and 1% penicillin-streptomycin for 5–7 days. BMDMs ($2 \times 10^5$ cells/mL) were seeded into 6-well culture plates. As for the animals' euthanasia practices, we anesthetized the rat with the intraperitoneal injection of ketamine (80 mg/kg) and xylazine, then the animals were sacrificed by cervical dislocation.

## Small interfering RNA (siRNA) transfection

siRNA targeting PAD4 (siPAD4) and scramble siRNA (siNC) were purchased from GenePharma incorporation (GenePharma, China). RASMCs, seeded in a 12-well plate at the density of $1 \times 10^6$ cells/well, were transfected with 50 nM siRNA using Lipofectamine iMAX reagent (13778030, Thermo Fisher, USA) for 8 h, then the subsequent related experiments were performed. The sequences of siRNAs were presented in Supplementary Table 3.

## Cells Cholesterol Loading and RASMC transdifferentiation experiment

Cholesterol-loading experiments were performed using water-soluble cholesterol from Sigma (C4951-30MG) as previously published[48]. All treatment concentrations involving Chol: MβCD were based on cholesterol weight (dissolved in 0.2% bovine serum albumin, BSA). Cells were allowed to grow to 70% confluency and then incubated with Chol/MβCD (10 μg/ml) for 72 h. After 72 h, cells were used for further experiments or harvested for mRNA and protein analysis.

## CD68⁺ VSMCs' ETs induction

RASMCs were incubated with Chol/MβCD for 72 h and transferred into a newly made medium containing 100 μg/ml ox-LDL (P00794, Solarbio). We harvested the cultivated supernatant for dsDNA detection, the fixed cells for IF staining, and the cells' protein for western blot (WB). The induction efficiency of CD68⁺ VSMCs' ETs was evaluated at different time points, and the maximal induction at the time point of 8 h.

## PAD4 inhibitor or DNase I intervention on CD68⁺ VSMCs producing ETs

After RASMCs were seeded in 12-well plates for 48 h, we initially induced transdifferentiation of RASMCs by loading cholesterol. Then, the cells were switched to a medium containing 100 μM Cl-amidine (S8141, Selleck) for 0.5 h to inhibit the activity of PAD4. The medium containing 100 μM BSA served as a control group. Subsequently, the treated and control groups were both stimulated by ox-LDL to induce ETs. As for the deoxyribonuclease DNase I experiment, we used 40 U/ml DNase I (11284932001, sigma) to pretreat the cells. Then cells were stimulated with ox-LDL.

## Real-time quantitative RT-PCR

Total RNA was extracted from cultured cells by using Trizol reagent (Thermo Fisher, USA) according to the manufacturer's instructions. Purified RNA (1000 ng) was reverse-transcribed by using PrimerScript RT Reagent Kit (Takara, Japan). Then, the quantitative RT-PCR was performed on 1 μg of cDNA product by using FastStart Universal SYBR Green Master (QR0100, Roche, USA) on a Roche Lightcycler (Version 1.1). Information on primers was presented in the Supplementary Table 1.

## Cell proliferation analysis and scratch assay

Cell proliferation was assessed by 5-ethynyl-2′-deoxyuridine (EDU) incorporation assay (C10337, Thermo, USA). The migratory rates of RASMCs pretreated with siPAD4 or siNC were assessed through scratch assay. After the cells reached 90% confluence, the wound was made by a straight scratch with a 200 μl sterile pipette tip across the center of the well. Then, cells were washed with PBS to remove the floating cells. Images were captured within 24 h after the scratch. The relative distance of cell migration was measured, and the percentage of healing was calculated.

## Immunofluorescence staining and laser confocal fluorescence microscopy analysis

Cells were seeded at $1 \times 10^5$ cells/well on glass-bottomed culture dishes and then fixed with 4% paraformaldehyde for 15 min, followed by permeabilizing with 0.2% Triton X-100 (X100RS, Sigma, USA) in PBS for 5 min. Sectioned aortic root, brachiocephalic arteries (BCAs), or human coronary plaques were fixed with paraformaldehyde for 15 min, followed by permeabilizing with 0.2% Triton X-100 in PBS for 5 min. Then, the permeabilized cells or tissues were washed with PBS three times and blocked with 5% goat serum. Finally, the sections or cells were incubated with antibodies diluted in blocking buffer, including anti-Citrullinated Histone 3 (ab5103, Abcam, 1:100), anti-α-smooth muscle actin (α-SMA, ab7817, Abcam, 1:200), anti-CD68 (NB100-683, Novus, 1:100), anti-LY6G (127601, Biolegend, 1:100), anti-MPO (AF3667, R&D, 1:100), anti-PAD4 (ab2148, Abcam, 1:100), anti-GSDMD (20770, Proteintech, 1:100), and anti-Ki-67 (ab15580, Abcam, 1:100). Normal isotype IgG (sc2027, Santa Cruz, 1:100) was used as the negative control. After three times of PBS wash, secondary antibodies Alexa Fluor 647-conjugated goat anti-rabbit (A-21244, Invitrogen, 1:200), Alexa Fluor 647-conjugated goat anti-rat (A-21247, Invitrogen, 1:200), Alexa Fluor 488-conjugated goat anti-rabbit (A-11008, Invitrogen, 1:200), Alexa Fluor 488-conjugated donkey anti-rat (A-21208, Invitrogen, 1:200), Alexa Fluor 594-conjugated goat anti-rat (A-11007, Invitrogen, 1:200), Alexa Fluor 594-conjugated donkey anti-rabbit (R37119, Invitrogen, 1:200), and Alexa Fluor 594-conjugated goat anti-mouse (A-11005, Invitrogen, 1:200) were incubated for 1 h at 37 °C in the dark. Nuclei were labeled with DAPI (Vector Laboratories), and cells were visualized using an LSM710 laser confocal microscope (Carl Zeiss,

Germany). A close examination of each plane of the z-stack was conducted using Zen 2009 Light Edition Software (Zeiss) to confirm that the presence of immunofluorescence staining coincided with a single DAPI nucleus. As for the assessment of CD68⁺ VSMCs within plaque lesion in the laser confocal fluorescence microscopy analysis, the double-positive region of α-SMA and CD68 was considered to be the region of CD68⁺ VSMCs according to the method of published research[49]. Image J was used to calculate the area of the lesion area or the positive region.

## Immunohistochemistry (IHC) and FISH analysis

For IHC, the antigen was repaired after the paraffin section was dewaxed. The sectioned tissues were blocked with 5% goat serum and then cultured with primary antibodies of MMP9 (ab38898, Abcam, 1:100) and CD68 (NB100-683, Novus, 1:100) at 37 °C for 1 h. After three times of PBS washes, the sections were incubated with secondary antibodies at 37 °C for 1 h. We used DAB Horseradish Peroxidase Color Development Kit (DA1015, Solarbio, China) to stain the sections. The cell nucleus was stained with Mayer's hematoxylin solution. Normal isotype-matched IgG (sc2025 or sc2026, Santa Cruz, USA) was used as the negative control. We used Image J to calculate the ratio of the positive area of MMP9 or CD68 in the area of total plaque lesion. FISH for scRNA-seq verification: sections were permeabilized with proteinase K for 5 min at 37 °C and treated with 3% $H_2O_2$ to block endogenous peroxidase activity. Slides were then incubated with the relevant probes at the concentration of 1 nM overnight in an airtight incubation chamber. Then, the signals were visualized using anti-digoxigenin-peroxidase and incubated with Cy3-tyramide in a blocking buffer; DAPI staining was used to visualize cell nuclei. A digoxigenin-labeled targeted mRNA (Klf4, Ccl2, Pad4, Col3a1, Vim, Acta2, and Spp1) oligo probes were purchased from Future Biotech (Beijing, China). Then we observed the results under the confocal microscope and calculated the ratio of positive Tdtomato⁺ cells.

## Multiplexed immunofluorescence

Multiplexed immunofluorescence (m-IF) was performed by sequentially staining frozen tissue sections with primary antibodies and paired with a TSA multiple-color kit (D110071-50T, WiSee Bio). For example, fixed frozen slides were incubated with anti-CD68 antibody (NB100-683, Novus, 1:100) for 30 min and then treated with anti-rat horseradish peroxidase-conjugated (HRP) secondary antibody for 10 min. IF labeling was developed for a strictly observed 10 min, using TSA 570 per manufacturer's direction. Between all steps, the slides were washed with Tris buffer. The same process was repeated for antibodies/fluorescent dyes: anti-H3CIT (ab5103, Abcam, 1:500)/TSA 670, anti-CD68 (NB100-683, Novus, 1:100)/TSA 570. Each slide was treated with 2 drops of DAPI, washed in PBS, and manually coverslipped. Slides were air dried, mounted with an Anti-fade mounting medium, and taken pictures with Aperio Versa 8 tissue imaging system (Leica). Images were analyzed using Image J software.

## Measurement of cell apoptosis

Measurement of apoptotic RASMCs was determined by flow cytometry-based Annexin V-PE staining (C1065S, Beyotime, China) and Hoechst (C1028, Beeyotime, China). The double-negative cells (viable), Annexin V single-positive cells (apoptosis), and double-positive cells (necrosis) were analyzed using FlowJo Software (V10.0.7, USA). Apoptotic cells in carotid arteries were assessed using the TUNEL assay (40307ES20, YEASEN, China). The number of TUNEL-positive cells was counted in 10 randomly selected fields in each sample under ×400 magnification by using Image J.

## Protein extraction and western blot

Whole cells from ex vivo experiments were prepared by 1× cell lysis buffer (9803, Cell Signaling Technologies) with protease inhibitors (04693159001, Roche Molecular Biochemicals, USA). Protein concentrations were determined by using a BCA protein assay. Proteins were separated by SDS–PAGE, transferred to polyvinylidene fluoride membranes, and incubated overnight at 4 °C with antibodies diluted with blocking liquid including anti-PAD4 (ab2148, Abcam, 1:800), anti-CD68 (NB100-683, Novus, 1:500), anti-α-SMA (ab7817, Abcam, 1:1000), anti-H3CIT (ab5103, Abcam, 1:500), anti-GAPDH (60004-1-Ig, Proteintech, 1:5000), anti-MYD88 (SC-74532, Santa Cruz, 1:600), anti-TLR4 (SC-10741, Santa Cruz, 1:500), anti-pSTAT3 (9145, Cell Signaling Technology, 1:1000), anti-Stat3 (9139, Cell Signaling Technology, 1:1000), anti-SOCS1 (ab9870, Abcam,1:800), and anti-STING (ab288157, Abcam, 1:1000). Then incubated with secondary antibody for 1 h and bends were visualized using chemiluminescence (TANON, China) and viewed under Amersham Imager 600 system (GE Healthcare, USA). Uncropped scans of the immunoblots are supplied in the Source Data file.

## Single-cell preparation of mouse atherosclerotic arterial specimens

Mice were sacrificed by cervical dislocation, and vasculature was flushed with PBS to completely remove blood. Arterial tissues, enriched with atherosclerosis plaque, including the aortic arch, ascending aorta, descending artery, brachiocephalic artery, thoracic aorta, and abdominal aorta, were isolated, minced to about 1 mm piece, and placed into tissue dissociation solution (130-110-201, Miltenyi Biotec). After incubation at 37 °C for 30 min in a magnetic stirrer, cell suspensions were filtered through 70 μm strainers and centrifuged at 400 × g for 5 min. The cell pellets were washed once with 0.2% FBS in PBS for FACS.

## Fluorescence-activated cell sorting (FACS)

Mouse aortic single cells for scRNA-seq were prepared from mice of *B6-G/R Myh11^Cre^Pad4^flox/flox* group or *B6-G/R Myh11^Cre^* group respectively (each group $n = 6$). All the mice were fed with HFD at the same time point of 24 weeks. After washing with FACS buffer, cell pellets were resuspended in FACS buffer with live/dead fixable viability stain 780 (565388, BD Horizon). Cells were gated on forward/side scatter parameters to exclude small debris and on forward scatter height versus the forward scatter area to exclude doublets. Dead cells (APC-Cy7⁺) were excluded. Live Tdtomato⁺ cells (VSMCs lineage cells) and ZsGreen1⁺ cells (No VSMCs-lineage cells) from the *B6-G/R Myh11^Cre^* group or *B6-G/R Myh11^Cre^Pad4^flox/flox* group ($n = 6$, respectively) were sorted separately via BD Influx instrument into two collection tubes and were immediately subjected to scRNA-seq using Chromium Single Cell Gene Expression system (10× Genomics) in parallel.

## Single-cell RNA sequencing

All scRNA-seq experiments were performed at the Single Cell Core Facility of OE Biotech Co. Ltd. (Shanghai, China). Samples were prepared using the 10×Genomics Chromium Single Cell 3' Reagent Kits v2 (mouse samples) according to the manufacturer's instructions. 5000 cells and 200 M reads per sample were targeted. 12 cycles for cDNA amplification and 12 cycles for sample index PCR were used. Sequencing was performed on a NovaSeq 6000 (R1 = 26 or 28, index = 8, R2 = 91).

## Analysis of mouse scRNA-seq data

The Cell Ranger software pipeline (version 5.0.0) provided by 10×Genomics was used to demultiplex cellular barcodes, map reads to the genome and transcriptome using the STAR aligner, and down-sample reads as required to generate normalized aggregate data across samples, producing a matrix of gene counts versus cells. The datasets were analyzed using Seurat v3.1.1 in R[50]. We filter out the cells with several expressed genes <200 or the percent of mitochondrial genes over 10% of total expressed genes. Furthermore, we remove the

potential doublets (and to an even lesser extent of higher order multiplet) that happen in the encapsulation step or as occasional pairs of cells that are not dissociated in the sample preparation step using the DoubletFinder package (version 2.0.2) of the R[51]. The following filters were applied to cells in each dataset: number of genes, number of UMIs, and the percentage of reads that mapped to mitochondrial genes. Raw UMI counts from each cell were normalized by the total counts in that cell, multiplied by 10,000, and then natural log transformed by adding a pseudo-count of 1. Highly variable genes were identified in each dataset based on mean expression and dispersion. Genes that have mean expression between 0.05 and 10 and dispersion between 1.5 and 20 were selected as highly variable genes. To integrate datasets across four samples including SMC lineages and Non-SMC lineages from *B6-G/R Myh11^cre* group and *B6-G/R Myh11^cre Pad4^flox/flox* group respectively while minimizing batch effect, we utilized the integration pipeline in Seurat built on canonical correlation analysis. To remove batch effect across samples, we performed integration analysis implemented in Seurat v3.1.1. Integration anchors were identified based on the union of highly variable genes across datasets (maximum of 1500 genes used in total) and the first 20 dimensions from the canonical correlation analysis. Integration was then performed using the anchors identified. Two-dimensional UMAP was used for visualization using the first 20 principal components built on the integration assay. Graph-based clustering was performed on the integrated dataset. In scRNA-seq analysis, a Shared Nearest Neighbor (SNN) graph was constructed with 50 nearest neighbors and 20 dimensions of PCs as input. Clusters were identified using the above graph with a resolution parameter of 0.4. Differential gene expression between clusters were performed using MAST test implemented in Seurat with each gene required to be expressed in at least 25% of cells in either of the two groups. Genes with $|\log_2 \text{foldchange}| > 0.5$ was set as the threshold for significantly differential expression. Bonferroni corrected $P$-value < 0.05 were considered differentially expressed.

### Pseudo-time trajectory analysis

Single-cell pseudo-time trajectories were reconstructed using the R package Monocle 3[52]. Dimensionality reduction was first performed with the DDRTree algorithm, using the expression of all highly variable genes detected as described above in 'scRNA-seq analysis'. As in scRNA-seq clustering analyses, to subtract confounding variation, cellular gene detection rate, percentage of mitochondrial gene expression, cell cycle scores and donor/batch effects were included as covariates in the residual model formula to subtract these effects from the data. Cell trajectory was then captured using the orderCells function, with the starting pseudotime state denoted as the end of the trajectory.

### GO and pathway enrichment analysis

Gene Ontology (GO) and pathway enrichment analyses were performed using the cluster Profiler R package (V4.0)[53]. The annotation Dbi R package org.Mm.eg.db was used to map gene identifiers. Cluster marker sets and differentially expressed genes were tested individually for overrepresentation, with all expressed/detected genes in each case used as a background control. In each case, GO gene sets were tested for overrepresentation in cluster markers or differentially expressed genes by computation of enrichment P values (the enricher R function, default parameters) from the hypergeometric distribution of total genes in the background gene set, the number of genes within background annotated with the gene set, the size of the gene set and the number of genes within the cluster marker/differentially expressed genes list annotated with the gene set. Hypergeometric P values were adjusted in each case for multiple testing using Benjamini–Hochberg correction as before[54]. The results were visualized as bar plots using the R packages clusterProfiler (4.0), enrichPlot (1.0) and ggplot2 (4.0)[55]. To score individual cells for pathway activities, by using the R package AUCell (2.0). Firstly, for each cell we used an expression matrix to compute gene expression rankings with the AUCell_buildRankings function and default parameters. The canonical pathway database was downloaded from the Broad Institute website, and canonical pathway gene sets were then used to score each cell where, for each gene set and cell, area-under-the-curve (AUC) values were computed (AUCell_calcAUC function) based on gene expression rankings, where AUC values represent the fraction of genes within the top-ranking genes for each cell that are defined as part of the pathway gene set.

### Bulk RNA-Seq and Bioinformatics Analysis

RNA was extracted from RASMCs of H3CIT + dsDNA stimuli group or control group by using the TRIzol™ reagent (15596018, Invitrogen) according to the manufacturer's instructions. The using concentration of H3CIT (32571, Cayman) and dsDNA (S1089, BD) was 0.03 µM and 300 ng/ml respectively. RNA integrity was assessed using the Agilent 2100 Bioanalyzer (Agilent Technologies, Santa Clara). Then the libraries were constructed using TruSeq Stranded mRNA LT Sample Prep Kit (Illumina, San Diego, CA, USA) according to the manufacturer's instructions. The transcriptome sequencing and analysis were conducted by OE Biotech Co. Ltd. (Shanghai, China). FASTQ sequence data were mapped to rat reference genome NCBI Rnor6.0 (https://www.ncbi.nlm.nih.gov/assembly/GCF_000001895.5/) with TopHat v2.1.1. The reads per gene were counted by using "HTseq" (https://htseq.readthedocs.io, version 0.6.0). Sample quality metrics and raw read counts were imported into R for further processing. The DESeq2 R package[56] was used to estimate library size factors, normalize counts and perform differential expression analyses. Benjamini–Hochberg multiple testing correction was used to compute FDR, and genes were considered significantly differentially expressed at <5% FDR. Principal component analysis was performed in R using the top 1000 most variable genes, with normalized DESeq2 variance-stabilizing transformation expression as input.

### Tfs predicted analysis

As for the Tfs regulated up-regulated DEGs of sc-seq, we utilize website KnockTF (http://www.licpathway.net/KnockTF/index.html). KnockTF provides a large number of available resources of human gene expression profile datasets associated with TFs regulated mRNA and relatated annotaion. We submit the DEGs screened into website and predicted the TFs with standard of regulated genes >100 and adjust $P$-value < 0.05.

### GSEA analysis

The gene set enrichment analysis created a list of all DEGs screened in scRNA-seq data. The reference gene sets selected GOc5.all.v7.1.symbols.gmt.(http://www.gsea-msigdb.org/gsea/msigdb/human/collections.jsp#C5). Normalized enrichment scores were acquired using gene set permutations 1000 times, and a cutoff P-value of 0.05 was used to filter the significant enrichment results. And the GSEA was performed by using the R package clusterProfiler (V4.0).

### Flow cytometry analysis of atherosclerosis aorta

The above suspensions cells from mouse atherosclerotic arterial specimens were stained with live/dead fixable viability stain 780 (565388, BD Horizon) and extracellular DNA dye (SYTOX Blue, S11348, Invitrogen) for 8 min and 10 min, respectively. Cell suspensions were blocked with CD16/32 antibody (101320, Biolegend, 1:100) for 15 min and then stained with corresponding fluorescently labeled antibodies, anti-LY6G (127641, Biolegend, 1:100), anti-CD68 (137016, Biolegend, 1:100), and anti-α-SMA (ab208844, Abcam, 1:100).

### Detection of plasmas dsDNA of mice and cell-cultivated supernatant

Blood was harvested from our atherosclerosis model mice and centrifuged at 800 g for 10 min to harvest the serum. After ETs induced

from CD68[+] VSMCs in vitro, the cultivated supernatant was collected, and then dsDNA concentration was measured. The concentration of dsDNA was routinely measured by an established available PicoGreen dsDNA Quantitation assay kit according to the provided protocols (P7589, Thermo Fisher).

### Statistical analysis
Data were shown as mean ± standard deviation. Two-side, unpaired Student's *t*-test was utilized to analyze the difference between two groups accompanied by normally distributed variables. Mann–Whitney *U*-test was used in non-normally distributed variables. Differences across three or more groups were tested by using one-way ANOVA followed by a post hoc analysis with the Bonferroni test. *P*-value < 0.05 was defined as statistical significance. Statistical analyses were done using GraphPad Prism (version 8.0).

### Reporting summary
Further information on research design is available in the Nature Portfolio Reporting Summary linked to this article.

## Data availability
FASTQ files and expression matrices from mouse scRNA-seq data or rat RNA-seq data are available from the NCBI Gene Expression Omnibus (GEO) database under the accession number GSE197073 and GSE197074 respectively. The source data underlying all Figures and Supplementary Figures are provided as a Source Data file. Source data are provided with this paper.

## Code availability
All the codes supporting the results of the bioinformatic analysis are available in the Github according to the link (https://github.com/ZhaiMing0/Extracellular-traps-from-activated-VSMCs).

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

## Acknowledgements

We thank the patients who participated in this study. This study was supported by grants No. 82070230[Peng], 91939101[Peng], 81670746[Jian] from the Chinese National Natural Science Foundation and grant No. SHDC2020CR4019 [Peng] from Clinical Research Plan of SHDC.

## Author contributions

M.Z. performed the animal experiments, flow cytometry, conducted immunoblots, performed bioinformatic analysis, analyzed data, and interpreted the results. S.G. performed the histology of mice artery samples and analyzed and interpreted the results. P.L. conducted cell culture, immunoblots, and immunofluorescence staining. Y.S. and W.K. performed the animal experiments. Q.Y., J.S., G.Y., and J.H. performed immunoblots and quantitative PCR. Y.Z. drew the mechanism diagram. M.W.F. helped interpret the data and wrote the manuscript. J.Z. and W.J. interpreted the results and wrote the manuscript. W.P. conceived the project, designed experiments, analyzed data, interpreted results, and wrote the manuscript.

## Competing interests

The authors declare no competing interests.
