## [Peer Review File · Nature Communications]

Extracellular traps from activated vascular smooth muscle cells
drive the progression of atherosclerosisREVIEWER COMMENTS

Reviewer #1 (Remarks to the Author):

Title: Macrophage extracellular traps originated from vascular smooth muscle cells play key roles in atherosclerosis progression

Summary:

In this manuscript, Zhai et al tested the hypothesis that SMC-derived macrophages secrete the equivalent of macrophage-derived extracellular traps (ET) and that this exacerbates development of in atherosclerotic plaques. Using histology, SMC-lineage tracing, scRNAseq, and in vitro SMC and macrophage models, the authors present a lot of data of interest. The in vitro models are well validated and certainly support the mechanism proposed by the authors. The in vivo models would benefit significantly by authors focusing on analysis of lesions in the brachiocephalic artery, as opposed to the aortic root, due to the resemblance of brachiocephalic artery lesions with human lesions. Also, humans do not develop aortic root lesions. The use of scRNAseq analysis is helpful, though the lack of sample replicates and unclear quality control make any quantitative comparisons between groups impossible. Finally, the identification of SMC-derived macrophages has been controversial, and due to the technical limitations of flow cytometry and spectral overlap of lineage tracing markers with autofluorescence of lipid-laden macrophages (see Bulut et al. ATVB. 2020), macrophages have been previously misidentified as SMC-derived. The authors provide excellent evidence for the ability of SMC to acquire macrophage-associated markers, but it is inappropriate to assume that the primary function of macrophages is shared by these cells, namely that of phagocytosis and local pathogen clearance. Without such additional evidence, it is recommended that the authors adjust their wording and refer to the cells they are studying as “SMC-derived macrophage-like cells”, or something similar to indicate the limitations of the results. They finally acknowledge this point in the Discussion but elsewhere refer to them as SMC-derived macrophages.

Positive Comments:

1. The authors did well in validating and characterizing their rat aortic smooth muscle cell isolates. This lends a lot of confidence in the findings from their in vitro model as being truly relevant to SMC found in vivo.
2. The authors' use of the Myh11-CreERT2 system for lineage tracing in combination with scRNAseq analysis is powerful and allows for the rigorous identification of SMC-derived phenotypic subpopulations.

Major Concerns:

1. On lines 393 to 395, the authors mention that they identified “SMC-derived macrophages that produced ETs by staining the macrophage marker CD68 and ETs marker H3CIT on 2 adjacent aorta plaque sections”. It is unclear how markers can be correlated between different cells in serial histological sections, unless the authors are claiming that these sections contain the same cells that have been split apart. This type of analysis was also used in Figure 5H, decreasing confidence in the interpretation of these results.

2. In Figure 3E, the Western blot for CD68 looks like a broad smear across the 4 loaded samples. This does not lend confidence that the stain worked, or it was performed on different samples. Note that the Acta2 (a-SMA) and GAPDH stains seemed to work just fine. The CD68 stain would be expected to look like these in terms of having a single discrete band representing a sample.

3. In lines 418-419, the authors state that their data “suggested that the dedifferentiated SMCs expressed little PAD4 unless transdifferentiating to macrophages under cholesterol loading”. This statement is not supported by their experiments given that they have not defined their model of “dedifferentiated” SMCs, instead using their control treated SMCs as a model of differentiated SMCs (due to expression of SMC-marker genes such as Myh11, Acta2, Cnn1, etc). Furthermore, in Supplementary Figure 3, the authors measure relative expression of Gal, Sca-1, and Oct4 in their RASMCs, but it is unclear what these genes are normalized to. Furthermore, it is unclear how the authors can claim that control SMC have 2-3 fold higher expression of Oct4 compared to Gal or Sca-1, considering that they do not provide data measuring absolute transcript numbers, nor do they

provide the transcriptional efficiency achieved among their various primer pairs for qPCR.

4. The statement in lines 419-421 claim that the expression levels of PAD4 in SMCs treated with cholesterol are approximately the same as those seen in BMDMs. This comparison can only be achieved if one quantified the expression of PAD4 (either transcript or protein) on a per-cell basis. As is, the authors make the implicit assumption that their loading control for their Western blot, GAPDH, is expressed at the same level per cell in RASMCs as it is in BMDMs.

5. In Figure 6, the authors present data from the scRNAseq analysis of TdTomato+ (SMC-derived) and zsGreen+ (non-SMC-derived) cells isolated from 24-week high fat diet-fed atherosclerotic mice. The clusters shown between the two are largely the same, with differences shown only in the relative proportions of each cluster (compare B6-G/R Myh11cre between Figure 6C and Figure 6D). Why are cells identified as Macrophages, Fibroblasts, and Endothelial Cells found in high relative abundance in the tdTomato+ population? This indicates that there was a technical issue (e.g. the FACS did not work as intended to separate the tdTomato and zsGreen populations).

6. In Figure 6 and 7, the authors are presenting scRNA-seq analysis of their atherosclerotic SMC-lineage tracing mice. Within this analysis the authors claim that their Pad4-deletion in SMC results in significant population changes within lesions. In order to make such a conclusion, it is necessary to assess the level of variance (i.e. standard deviation of relative population size) within each subpopulation found, thus requiring replicate scRNA-seq samples. Although this may be cost prohibitive, the assessment of variance is absolutely necessary to claim any population changes are occurring.

Minor Concerns:

1. Typographical errors and unclear grammar are prevalent throughout the manuscript, often making it difficult to interpret statements.

2. The authors state in lines 666-667 that "macrophages (myeloid and non-myeloid subtypes) are the main inflammatory cells present". This is a confusing statement because macrophages, by definition, are myeloid-derived. I think the authors meant to state that SMC-derived macrophage-like cells, in addition to regular myeloid-derived macrophages, are the main inflammatory cells present in the lesions. Please clarify this statement.

Reviewer #2 (Remarks to the Author):

First of all, it is a more than extensive work. However, it needs a major revision, especially as far as the form of presentation and the dates and figures are concerned.

major

1. The manuscript is far too long and contains too many figures. That makes reading and reviewing very difficult. The main text has to be shortened considerably (approximately 50%) and made concise. And all subfigures have to be tested for importance for the message of the manuscript. All figures that are just supportive have to be transferred to the supportive materials. All subfigures that are not required at all have to be skipped.

2. In my understanding a macrophage is a hematopoietic cell somehow derived from the bone marrow. If a VSMC expresses CD68 or other markers that are usually expressed by macrophages it is not a macrophage. Therefore, the wording of the text has to be modified. I propose to name the cells CD68-positive VSMC or VSMC68.

3. I do not see data to conclude that the VSMC68 are phagocytic, however, this is not important if the word macrophage is eliminated.

4. My proposal for the title is "Extracellular traps from activated vascular smooth muscle cells drive the progression of atherosclerosis". If you can show that the ET forming cells are really related to VSMC.

5. I do not doubt that the authors have seen NETs when they inspected their figures on a big screen. However, I cannot see NETs in the figures they present. Nor do I see data that show that the NETs are derived from cell that had been VSMC before. If the sialoadhesin in the lesions is derived from

“transdifferentiated” VSMC or from necrotic canonical macrophages is not clear. High definition figures have to be included to support this message.

6. I also do not see data that show that the ETs are derived from macrophage-like cells that were originated from VSMC.

7. The figures 1D 1E 2A 2E 2F 4H, 4I, 5H, 8M, 8N, supp2E, supp3A, supp3B, supp3E, supp3F, supp4E, supp6F, supp7E, have to be displayed like 3O. i.e. with single fluorescence and merge in the supportive material. Otherwise, the figures are not understandable.

8. Figure 3N has to be edited and the interception has to be removed

9. Remove figure 6ABEF, figure 7 and figure 8 into the supportive materials; Just keep Figure 6CD in the paper.

minor:

1. The English requires proofreading, preferentially by a native speaker.

2. Do not use NETosis or Tosis but NET formation or formation of extracellular DNA traps.

Martin Herrmann

Reviewer #3 (Remarks to the Author):

In this study, the authors focus on a new biological process of SMC-derived macrophages generating extracellular traps (ETs) and its function in atherosclerosis. By utilizing SMC-lineage tracing technology and scRNA-seq, they demonstrate that the ETs from SMC-derived macrophages influence the progress of atherosclerosis.

Key finding of this study, ‘SMC-derived macrophages influence the progress of atherosclerosis by regulating the direction of SMC’s trans-differentiation through activating STING-SOCS1 or TLR4 signaling pathway of surrounding SMCs’ is of interest as not much is known about the macrophage generated ETs. However, there are some concerns in this study that need to be addressed. They are

1. In multiple places, it is difficult to clearly understand what the authors want to convey. Here a few examples:

“Moreover, we detected mRNA expressed levels of PAD family members and stem-related genes between BSA-treated RASMCs and RASMCs loaded with cholesterol for 72 hours”.

“Macrophages were divided into three subtypes, while PAD4(hi) Macrophages and Macrophage I expressed higher levels of pro-inflammation related genes compared with Macrophage II such as Spp1, Ccl3 and Cxcl2 “.

“The detail was illustrated in supplementary files”

2. What are the quality control parameters used in filtering out low quality cells before clustering using Seurat?

3. What was the reason behind using ‘MAST’ for differential expression analysis instead of the default ‘wilcox’ method?

4. Seurat developers claim that their RPCA integration algorithm works better when studying datasets with subtle biological differences. CCA could over-integrate the cells. Have you checked your analysis using other integration methods to see if you get similar results?

A point-by-point response to the Reviewers' critique

We thank the three Reviewers for the valuable and constructive suggestions on our manuscript. Please find our point-by-point response below. All changes and revisions in the edited manuscript are highlighted by yellow text color for tracking purposes. New figure panels and supplementary figures are indicated by yellow-color figure legends.

Reviewer #1:

Summary:

In this manuscript, Zhai et al. tested the hypothesis that SMC-derived macrophages secrete the equivalent of macrophage-derived extracellular traps (ET) and that this exacerbates the development of in atherosclerotic plaques. Using histology, SMC-lineage tracing, scRNAseq, and in vitro SMC and macrophage models, the authors present a lot of data of interest. The in vitro models are well validated and certainly support the mechanism proposed by the authors. The in vivo models would benefit significantly by authors focusing on analysis of lesions in the brachiocephalic artery, as opposed to the aortic root, due to the resemblance of brachiocephalic artery lesions with human lesions. Also, humans do not develop aortic root lesions. The use of scRNAseq analysis is helpful, though the lack of sample replicates and unclear quality control make any quantitative comparisons between groups impossible. Finally, the identification of SMC-derived macrophages has been controversial, and due to the technical limitations of flow cytometry and spectral overlap of lineage tracing markers with autofluorescence of lipid-laden macrophages (see Bulut et al. ATVB. 2020), macrophages have been previously misidentified as SMC-derived. The authors provide excellent evidence for the ability of SMC to acquire macrophage-associated markers, but it is inappropriate to assume that the primary function of macrophages is shared by these cells, namely that of phagocytosis and local pathogen clearance. Without such additional evidence, it is recommended that the authors adjust their wording and refer to the cells they are studying as “SMC-derived macrophage-like cells”, or something similar to indicate the limitations of the results. They finally acknowledge this point in the Discussion but elsewhere refer to them as SMC-derived macrophages.

Positive Comments:

- 1. The authors did well in validating and characterizing their rat aortic smooth muscle cell isolates. This lends a lot of confidence in the findings from their in vitro model as being truly relevant to SMC found in vivo.*
- 2. The authors' use of the Myh11-CreERT2 system for lineage tracing in combination with scRNAseq analysis is powerful and allows for the rigorous identification of SMC-derived phenotypic subpopulations.*

Response: We thank the Reviewer for the positive comments on our manuscript and the valuable suggestions.

Concerns:

Major Concerns:

- 1. On lines 393 to 395, the authors mention that they identified “SMC-derived macrophages that*

produced ETs by staining the macrophage marker CD68 and ETs marker H3CIT on 2 adjacent aorta plaque sections”. It is unclear how markers can be correlated between different cells in serial histological sections unless the authors are claiming that these sections contain the same cells that have been split apart. This type of analysis was also used in Figure 5H, decreasing confidence in the interpretation of these results.

Response: Thanks very much for the Reviewer’s valuable comments. In the previous manuscript version, we identify VSMCs-derived macrophages (We change their name to “CD68⁺ VSMCs” according to the suggestion of reviewer 2 in the new manuscript) that produce ETs by staining different markers on the two adjacent aorta plaque sections and find ETs production is decreased in B6-G/R *Myh11^{Cre} Pad4^{flx/flx}* group compared with the control group. The thickness of the pathological sections is 5µm, approximately reflecting the co-staining area of the CD68 marker and H3CIT marker, which is similar to the method used in a previous study to locate specific areas containing NETosis neutrophils in cross sections¹ (e.g., Libby P et al. Roles of PAD4 and NETosis in Experimental Atherosclerosis and Arterial Injury: Implications for Superficial Erosion. *Circ Res.* 2018). To address the Reviewer’s concern, we adopt multiple IF (m-IF) technology, and it allows us to observe ETs produced from VSMCs-derived macrophages on the one plaque section from VSMCs lineage tracing mice. We perform the m-IHC in the lesion of the aortic root of B6-G/R *Myh11^{Cre}* mice, confirming the previously obtained data that ETs produced from CD68⁺ VSMCs located at the fibrous cap or shoulder position of the plaque. We have updated the figure in the **new Figure 1H**. Meanwhile, we also replace previous results with the m-IF results in the **new Figure 4E&4F** between 2 groups, and we obtain similar results that ETs indeed diminish dramatically in the B6-G/R *Myh11^{Cre} Pad4^{flx/flx}* group, and the ratio of VSMCs derived macrophages decreased.

2. In Figure 3E, the Western blot for CD68 looks like a broad smear across the 4 loaded samples. This does not lend confidence that the stain worked, or that it was performed on different samples. Note that the Acta2 (α-SMA) and GAPDH stains seemed to work just fine. The CD68 stain would be expected to look like these in terms of having a single discrete band representing a sample.

Response: Thanks very much for the Reviewer’s valuable suggestion. To address the concern of smear results in the Figure 3E of the previous manuscript, we repeat the WB experiment between the RASMCs control group and RASMCs cholesterol stimulated group and replace it with a clear result in **new Figure 2E-G**, having a single discrete band representing a sample.

3. In lines 418-419, the authors state that their data “suggested that the dedifferentiated SMCs expressed little PAD4 unless transdifferentiating to macrophages under cholesterol loading”. This statement is not supported by their experiments given that they have not defined their model of “dedifferentiated” SMCs, instead using their control treated SMCs as a model of differentiated SMCs (due to expression of SMC-marker genes such as *Myh11*, *Acta2*, *Cnn1*, etc). Furthermore, in Supplementary Figure 3, the authors measure the relative expression of *Gal*, *Sca-1*, and *Oct4* in their RASMCs, but it is unclear what these genes are normalized to. Furthermore, it is unclear how the authors can claim that control SMC has 2-3-fold higher expression of *Oct4* compared to *Gal* or *Sca-1*, considering that they do not provide data measuring absolute transcript numbers, nor do they provide the transcriptional efficiency achieved among their various primer pairs for qPCR.

Response: We thank the Reviewer for raising this critical point. Control-treated RASMCs do not represent de-differentiated VSMCs, although isolated from rat aorta. We use PDGF to induce RASMCs’ de-differentiation and detect the expression of VSMCs-marker genes, such as *Myh11*, *Acta2*, and *Cnn1*. After being stimulated with PDGF, the expression of VSMCs-marker genes is decreased. Then, the stimulated VSMCs are chosen as a de-differentiated VSMCs model. Stem-cell-related genes such as *OCT4*, *Sca-1*, and *Gal* are down-regulated in trans-differentiated VSMCs (cholesterol loading) compared with de-differentiated VSMCs. The data is shown in the new Supplementary Figure 3C. As for the concern that ‘control VSMCs have 2-3-fold higher expression of *Oct4* compared with *Gal* or *Sca-1*’, we are sorry that our incorrect illustration confused the reader. We don’t mean that *Oct4* is 2-3-fold higher in control VSMCs compared with *Gal* and *Sca-1*; we want to illustrate that PAD4 upregulation is most obviously among PAD family in trans-differentiated VSMCs compared with control. Moreover, we compare the mRNA expression of the PADs family between trans-differentiated VSMCs and VSMCs or between de-differentiated VSMCs and VSMCs, and we find that only PAD4 elevated significantly among PADs family unless VSMCs are loaded with cholesterol. The data is shown in the new Supplementary Figure 3D. The relevant statement has been revised in the manuscript from line 372 to line 377.

4. The statement in lines 419-421 claims that the expression levels of PAD4 in SMCs treated with

cholesterol are approximately the same as those seen in BMDMs. This comparison can only be achieved if one quantified the expression of PAD4 (either transcript or protein) on a per-cell basis. As is, the authors make the implicit assumption that their loading control for their Western blot, GAPDH, is expressed at the same level per cell in RASMCs as it is in BMDMs.

Response: We thank the reviewer for their carefulness. Indeed, in the previous manuscript, we could not guarantee that the GAPDH is expressed at the same level per cell in RASMCs as it is in BMDMs because we ignore different intrinsic natures of cells. We search the GAPDH expression on the website of Protein Atlas (<https://www.proteinatlas.org>) and find that the RNA expression in macrophages and VSMC is 2134 vs. 2878.8 nTPM, respectively (**Reviewer-only Figure 1A**). Moreover, to address this concern, we keep the number of cells between two groups approximately the same by cell count, then we repeat experiments at transcript or protein. We find that after cell count control, the PAD4's expression of cholesterol-loading RASMCs is not as much as that of BMDMs either at mRNA (**Reviewer-only Figure 1B**) or protein level (**new Figure 2I&J**). But, PAD4 indeed express highly in challenged RASMCs compared with control RASMCs. We have deleted “the expression level was nearly similar to BMDMs used as positive controls” in lines 378-379 of the new manuscript and replace it with a new figure in **new Figure 2I&J**.

Reviewer-only Figure 1: A, The mRNA expression level of GAPDH within single cells between Smooth muscle cells and Macrophages according to the results of Protein Atlas. B, the relative mRNA expression of PAD4 among RASMCs, cholesterol-treating RASMCs, and BMDM.

5. In Figure 6, the authors present data from the scRNAseq analysis of tdTomato⁺ (SMC-derived) and zsGreen⁺ (non-SMC-derived) cells isolated from 24-week high fat diet-fed atherosclerotic mice. The clusters shown between the two are largely the same, with differences shown only in the relative proportions of each cluster (compare B6-G/R Myh11^{cre} between Figure 6C and Figure 6D). Why are cells identified as Macrophages, Fibroblasts, and Endothelial Cells found in high relative abundance in the tdTomato⁺ population? This indicates that there was a technical

issue (e.g. the FACS did not work as intended to separate the tdTomato and zsGreen populations).

Response: Thanks for the Reviewer's valuable and helpful suggestions. We identified several types of cells within Tdtomato⁺ cells, such as Macrophages, Fibroblasts, and Endothelial Cells based on their differentially expressed markers from our scRNA-seq results. For example, the macrophages derived from VSMCs expressed *CD68*, fibroblasts derived from VSMCs expressed *Vim* and *Tnc*, and Endothelial cells derived from VSMCs expressed *CD31* and *Cdh5*, which shows in the new Supplementary Figures 6C & 6E-F. Our results of scRNA-seq combined with VSMCs-lineage tracing work are consistent with previous papers²⁻⁴, which also find several VSMCs derived types of cells (such as Macrophages, SEM cells, fibroblasts, and fibro-chondrocyte). Our FACS experiment is conducted based on the apparent gating strategy, and groups of Tdtomato⁺ cells or ZsGreen⁺ cells distinguished well, then they are sent for scRNA-seq, shown in the new Supplementary Figure 6B. mRNA fluorescence in situ hybridization (mFISH) is often used to verify new cell types by single cell sequencing⁵⁻⁷, and to provide information on spatial locations of cell types and cell ratio changes between groups. Furthermore, to address the Reviewers' concerns, we use lineage tracing technology and mFISH to extinguish these types of cells derived from VSMCs that express specifically identified mRNA (in the new Supplementary Figure 8).

6. In Figures 6 and 7, the authors are presenting a scRNA-seq analysis of their atherosclerotic SMC-lineage tracing mice. Within this analysis, the authors claim that their Pad4-deletion in SMC results in significant population changes within lesions. To make such a conclusion, it is necessary to assess the level of variance (i.e. standard deviation of relative population size) within each subpopulation found, thus requiring to replicate scRNA-seq samples. Although this may be cost prohibitive, the assessment of variance is absolutely necessary to claim any population changes are occurring.

Response: We thank you for the Reviewers' rigorous and meticulous work. In this study, we used a total of 54 PAD4 VSMCs conditional knockout mice and control mice, with 27 mice in each group for detection of plaque load, pathology, molecular and other indicators. In addition, after the successful mouse modeling test, we randomly selected six representative model mice from each group. Then, we performed scRNA-seq of pooled Tdtomato⁺ cells and ZsGreen⁺ cells sorted from atherosclerotic aortas of B6-G/R *Myh11^{Cre}Pad4^{lox/lox}* mice and B6-G/R *Myh11^{Cre}* mice (A total of 12 littermate mice, each group is 6 mice were used in this experiment). We emphasize that the number of mice used in the experimental verification is more than 3. Therefore, our data are in line with the research requirements. In addition, there are many single-cell sequencing studies with less than 3 samples⁷⁻¹⁰, mainly through experimental validation to reveal scientific questions, which is a common practice in single-cell sequencing. Indeed, it is necessary to assess the level of variance (i.e., the standard deviation of relative population size) within each subpopulation found. Thus, to address the major concerns of the Reviewer, we performed mFISH (N=3) in the atherosclerosis plaque of VSMCs lineage tracing mouse between 2 groups to verify the ratio changes of cell populations identified by our scRNA-seq bioinformatics analysis (new Supplementary Figure 8). mFISH is a powerful indicator for routine detection of scRNA-seq analysis results. The mFISH probe (*Klf4*, *Ccl2*, *Pad4*, *Col3a1*, *Vim*, *Acta2*, and *Spp1*) is in line with the markers used to distinguish different cell types of scRNA-seq analysis (mFISH probe of *Klf4* is used to identify Stem-Like SMC, *Ccl2* is used to distinguish senescent SMC, *PAD4* is used to distinguish PAD4(hi) CD68⁺ SMC, *Col3a1* is used to distinguish Fibro-chondrocyte, *Vim* is used to distinguish Fibroblast, *Acta2* is used to distinguish VSMCs and *Spp1* is used to distinguish transdifferentiated harmful-like cells). We believe that our research is innovative and will excite readers.

Supplementary Figure 8

Minor Concerns:

1. Typographical errors and unclear grammar are prevalent throughout the manuscript, often making it difficult to interpret statements.

Response: Thanks for the Reviewer's suggestion. We agree with this suggestion and have appropriately modified the terminology throughout the text. In the specific case blow, such as in lines 361, 379, and 485 we have noted those and corrected them individually.

2. The authors state in lines 666-667 that “macrophages (myeloid and non-myeloid subtypes) are the main inflammatory cells present”. This is a confusing statement because macrophages, by definition, are myeloid-derived. I think the authors meant to state that SMC-derived macrophage-like cells, in addition to regular myeloid-derived macrophages, are the main inflammatory cells present in the lesions. Please clarify this statement.

Response: Thanks for the Reviewer’s suggestion. We have revised the statement related to VSMCs-derived macrophages in lines according to the Reviewer’s requirement. Please find our revision in lines 528 to 529.

Reviewer #2:

First of all, it is more than extensive work. However, it needs a major revision, especially as far as the form of presentation and the dates and figures are concerned.

Response: We thank the reviewer for the constructive suggestions and comments on this manuscript.

1. The manuscript is far too long and contains too many figures. That makes reading and reviewing very difficult. The main text has to be shortened considerably (approximately 50%) and made concise. And all subfigures have to be tested for importance for the message of the manuscript. All figures that are just supportive have to be transferred to the supportive materials. All subfigures that are not required at all have to be skipped.

Response: We agree with this suggestion and have shortened the main text and reformatted our figures. Some figures, just supportive, have been transferred to the supplementary materials. Meanwhile, some subfigures estimated not so as important were skipped according to the reviewer’s comments. For example, we have reduced the length of our text, the number of figures is reduced from 8 to 6 and other data were put in the supplementary figure.

2. In my understanding a macrophage is a hematopoietic cell somehow derived from the bone marrow. If a VSMC expresses CD68 or other markers that are usually expressed by macrophages it is not a macrophage. Therefore, the wording of the text has to be modified. I propose to name the cells CD68-positive VSMC or VSMC68.

Response: We thank the reviewer for pointing out this issue. We agree that the VSMCs express CD68 or other markers should be termed as CD68-positive VSMCs instead of VSMCs-derived macrophages, and we have modified it to “CD68 positive VSMCs (CD68⁺ VSMCs)” in our manuscript according to your advice.

3. I do not see data to conclude that the VSMC68 are phagocytic, however, this is not important if the word macrophage is eliminated.

Response: Thank you on this point. Gary K Owens et al. illustrated that cholesterol loading increase VSMCs’ phagocytic behavior¹¹. To address the reviewer’s concern, we detected phagocytic genes between the control group and cholesterol-treated group, and both are incubated with Red Zymosan Bioparticles (Invitrogen, P35364, USA). We find that cholesterol-loading VSMCs indeed highly expressed phagocytic genes and have more bioparticles phagocytosed, while no cholesterol-loading VSMCs have little bioparticles phagocytosed. It is worth noting that the cholesterol-loading VSMCs express less protein of α -SMA compared with no-cholesterol-loading VSMCs, which are shown in the **new Supplementary Figures 3A & 3B**. These results are consistent with the results previously reported¹¹.

To make our manuscript more rigorous, we change the term “VSMCs derived macrophages” to “CD68 positive VSMCs”, according to the reviewer’s suggestion.

4. My proposal for the title is “Extracellular traps from activated vascular smooth muscle cells drive the progression of atherosclerosis”. If you can show that the ET forming cells are related to VSMC.

Response: We agree with this proposal. Furthermore, we will be happy to edit the text further based on helpful comments from the reviewers. Furthermore, we repeat our experiments in vivo and in vitro to observe ET forming cells related to VSMCs, results performed following in response to comment 5 and comment 6 respectively.

5. I do not doubt that the authors have seen NETs when they inspected their figures on a big screen. However, I cannot see NETs in the figures they present. Nor do I see data that show that the NETs are derived from a cell that had been VSMC before. If the sialoadhesin in the lesions is derived from “transdifferentiated” VSMC or from necrotic canonical macrophages is not clear. High-definition figures have to be included to support this message.

Response: Actually, we detect CD68⁺ VSMCs produced ETs by staining the macrophage marker CD68 and ETs marker H3CIT on 2 adjacent aorta plaque sections, which may influence the confidence of our results. To get high-definition figures and address the reviewer’s concerns, we adopt multiple IF (m-IF) technology, which allows us to observe ETs produced from VSMCs-derived macrophages on the same plaque section harvested from VSMCs lineage tracing mice. Referenced with the previous methods that observed NETs in atherosclerosis plaque, we assess the extracellular traps generated from CD68⁺ VSMCs colocalizing diffuse DNA and H3CIT in CD68 positive VSMCs area¹. We indeed detect that extracellular traps (ETs) are derived from cells that have been VSMCs before within atherosclerosis plaque lesions, as shown in the new Figures 1H-I.

6. I also do not see data that show that the ETs are derived from macrophage-like cells that originated from VSMC.

Response: We agree that our IF staining results of ETs generating by CD68 positive VSMCs in vitro are unsatisfactory. To address the reviewer’s concerns, we repeat this experiment and got a high-quality image of ETs by staining with H3CIT, α -SMA, and CD68, observing by confocal microscope with high

resolution. We can watch those extracellular DNA fibers (blue), co-stained with H3CIT (green), extending beyond the CD68 positive RASMCs (red and yellow merged) by single fluorescence, merged images, and 3D-construction image, shown in new Figure 2O & Supplementary Figure 3E-H.

7. The figures 1D 1E 2A 2E 2F 4H, 4I, 5H, 8M, 8N, supp2E, supp3A, supp3B, supp3E, supp3F, supp4E, supp6F, supp7E, have to be displayed like 3O. i.e. with single fluorescence and merge in the supportive material. Otherwise, the figures are not understandable.

Response: We agree with this suggestion and apologize for the confusion attributed to the single fluorescence image's lack and put the single fluorescence and merged figures in related supportive figures. These are shown in new Supplementary Figure 1E&1J, new Supplementary Figure 2G-I, new Supplementary Figure 4F, new Supplementary Figure 9F, and new Supplementary Figure 10J&K, respectively.

8. Figure 3N has to be edited and the interception has to be removed

Response: We thank you for the reviewer's helpful suggestion and edited Figure 3N in new Figure 2N. The interception has been removed.

9. Remove figure 6ABEF, figure 7 and figure 8 into the supportive materials; Just keep Figure 6CD in the paper.

Response: We thank you for the reviewer's constructive comments. We have re-arranged our figures of scRNA-seq to make our manuscript more readable.

Minor:

1. The English require proofreading, preferentially by a native speaker.

Response: We apologize for the poor language of our manuscript. We have now worked on both language and readability and involved native English speakers in language corrections. We hope that the flow and language level have been substantially improved.

2. Do not use NETosis or Tosis but NET formation or formation of extracellular DNA traps.

Response: Thanks for the reviewer's helpful suggestion. We have modified the illustration in our manuscript.

Reviewer #3:

In this study, the authors focus on a new biological process of SMC-derived macrophages generating extracellular traps (ETs) and its function in atherosclerosis. By utilizing SMC-lineage tracing technology and scRNA-seq, they demonstrate that the ETs from SMC-derived macrophages influence the progress of atherosclerosis.

Key finding of this study, 'SMC-derived macrophages influence the progress of atherosclerosis by regulating the direction of SMC's trans-differentiation through activating STING-SOCS1 or TLR4 signaling pathway of surrounding SMCs' is of interest as not much is known about the macrophage generated ETs. However, there are some concerns in this study that need to be addressed.

Response: Thank you for the positive assessment of our manuscript and your helpful comments.

Concerns:

1. In multiple places, it is difficult to clearly understand what the authors want to convey. Here are few examples:

"Moreover, we detected mRNA expressed levels of PAD family members and stem-related genes between BSA-treated RASMCs and RASMCs loaded with cholesterol for 72 hours".

"Macrophages were divided into three subtypes, while PAD4(hi) Macrophages and Macrophage I expressed higher levels of pro-inflammation related genes compared with Macrophage II such as Spp1, Ccl3, and Cxcl2".

"The data was illustrated in supplementary files".

Response: We apologize for the confusion generated by the previous version of the manuscript and sincerely hope that the issues pointed out will be easier to follow in this new version. In the specific case blow, we have noted those individually. For example, in lines 376-377, we have deleted "BSA-treated RASMCs and RASMCs...", in lines 451-452, we delete "Macrophages were divided into three subtypes...", and in lines 313-314, we have modified "the detail was illustrated in supplementary files" to "A full description of bioinformatic analysis can be found in supplementary methods."

2. What are the quality control parameters used in filtering out low-quality cells before clustering using Seurat?

Response: Thanks for the reviewer's helpful and constructive comments. We filter out the cells with several expressed genes < 200 or the percent of mitochondrial genes over 10% of total expressed genes. Furthermore, we remove the potential doublets (and to an even lesser extent of higher order multiplet) that happen in the encapsulation step or as occasional pairs of cells that are not dissociated in the sample preparation step using the DoubletFinder package (version 2.0.2) of the R¹². We have added the control parameters in the method of scRNA-seq analysis in new supplementary methods.

3. What was the reason behind using 'MAST' for differential expression analysis instead of the default 'Wilcox' method?

Response: Thanks for the reviewer's comments. We conduct the differential expression analysis of our scRNA-seq data by using the 'MAST' attribute to the 'MAST' method cost less time, and the results are a little different from the method using 'Wilcox'¹². To address the reviewer's concerns, we perform differential expression analysis using the 'Wilcox method between groups and compare

the results with that of the previous version. The DEGs results are near the same (shown in **Reviewer-only Figure 2**), and the subsequent analysis results are consistent with that in the former version.

Reviewer-only Figure 2: the Venn diagram results of DEGs screened from the MAST test or Wilcox test, respectively.

4. Seurat developers claim that their RPCA integration algorithm works better when studying datasets with subtle biological differences. CCA could over-integrate the cells. Have you checked your analysis using other integration methods to see if you get similar results?

Response: We thank you for the reviewer’s careful and helpful comments on our manuscript. According to the reviewers’ suggestions, we perform the integration of scRNA-seq data by using the RPCA integration algorithm and compare the results with that in the previous version. The changes in cell ratio between groups are similar to our previous results (shown in **Reviewer-only Figure 3**).

Reviewer-only Figure 3: **A** shows the integration results by using RPCA integration. **B** shows the

integration results by using CCA results. **C&D** show the cell type atlas by using the RPCA method. **E** shows the Tdtomato⁺ cells' atlas by using the RPCA method. **F** shows a diagram of ratio change among 4 samples by using the RPCA method. **G** shows a diagram of ratio change between 2 groups of Tdtomato⁺ cells.

Reference:

1. Franck G, Mawson TL, Folco EJ, Molinaro R, Ruvkun V, Engelbertsen D, Liu X, Tesmenitsky Y, Shvartz E, Sukhova GK, Michel JB, Nicoletti A, Lichtman A, Wagner D, Croce KJ and Libby P. Roles of PAD4 and NETosis in Experimental Atherosclerosis and Arterial Injury: Implications for Superficial Erosion. *Circ Res.* 2018;123:33-42.
2. Pan H, Xue C, Auerbach BJ, Fan J, Bashore AC, Cui J, Yang DY, Trignano SB, Liu W, Shi J, Ihuegbu CO, Bush EC, Worley J, Vlahos L, Laise P, Solomon RA, Connolly ES, Califano A, Sims PA, Zhang H, Li M and Reilly MP. Single-Cell Genomics Reveals a Novel Cell State During Smooth Muscle Cell Phenotypic Switching and Potential Therapeutic Targets for Atherosclerosis in Mouse and Human. *Circulation.* 2020;142:2060-2075.
3. Wirka RC, Wagh D, Paik DT, Pjanic M, Nguyen T, Miller CL, Kundu R, Nagao M, Coller J, Koyano TK, Fong R, Woo YJ, Liu B, Montgomery SB, Wu JC, Zhu K, Chang R, Alamprese M, Tallquist MD, Kim JB and Quertermous T. Atheroprotective roles of smooth muscle cell phenotypic modulation and the TCF21 disease gene as revealed by single-cell analysis. *Nat Med.* 2019;25:1280-1289.
4. Chattopadhyay A, Kwartler CS, Kaw K, Li Y, Kaw A, Chen J, LeMaire SA, Shen YH and Milewicz DM. Cholesterol-Induced Phenotypic Modulation of Smooth Muscle Cells to Macrophage/Fibroblast-like Cells Is Driven by an Unfolded Protein Response. *Arterioscler Thromb Vasc Biol.* 2021;41:302-316.
5. Kelly NH, Huynh NPT and Guilak F. Single cell RNA-sequencing reveals cellular heterogeneity and trajectories of lineage specification during murine embryonic limb development. *Matrix Biol.* 2020;89:1-10.
6. Moffitt JR, Bambach-Mukku D, Eichhorn SW, Vaughn E, Shekhar K, Perez JD, Rubinstein ND, Hao J, Regev A, Dulac C and Zhuang X. Molecular, spatial, and functional single-cell profiling of the hypothalamic preoptic region. *Science.* 2018;362.
7. Zhang Y, Li S, Jin P, Shang T, Sun R, Lu L, Guo K, Liu J, Tong Y, Wang J, Liu S, Wang C, Kang Y, Zhu W, Wang Q, Zhang X, Yin F, Sun YE and Cui L. Dual functions of microRNA-17 in maintaining cartilage homeostasis and protection against osteoarthritis. *Nat Commun.* 2022;13:2447.
8. Hendrikx T, Porsch F, Kiss MG, Rajcic D, Papac-Miličević N, Hoebinger C, Göderle L, Hladik A, Shaw LE, Horstmann H, Knapp S, Derdak S, Bilban M, Heintz L, Krawczyk M, Paternostro R, Trauner M, Farlik M, Wolf D and Binder CJ. Soluble TREM2 levels reflect the recruitment and expansion of TREM2(+) macrophages that localize to fibrotic areas and limit NASH. *J Hepatol.* 2022.
9. Goodyer WR, Beyersdorf BM, Paik DT, Tian L, Li G, Buikema JW, Chirikian O, Choi S, Venkatraman S, Adams EL, Tessier-Lavigne M, Wu JC and Wu SM. Transcriptomic Profiling of the Developing Cardiac Conduction System at Single-Cell Resolution. *Circ Res.* 2019;125:379-397.
10. McCann JV, Xiao L, Kim DJ, Khan OF, Kowalski PS, Anderson DG, Pecot CV, Azam SH, Parker JS, Tsai YS, Wolberg AS, Turner SD, Tatsumi K, Mackman N and Dudley AC. Endothelial miR-30c suppresses tumor growth via inhibition of TGF- β -induced Serpine1. *J Clin Invest.* 2019;129:1654-1670.
11. Shankman LS, Gomez D, Cherepanova OA, Salmon M, Alencar GF, Haskins RM, Swiatlowska P, Newman AA, Greene ES, Straub AC, Isakson B, Randolph GJ and Owens GK. KLF4-dependent phenotypic modulation of smooth muscle cells has a key role in atherosclerotic plaque pathogenesis. *Nat Med.* 2015;21:628-37.
12. McGinnis CS, Murrow LM and Gartner ZJ. DoubletFinder: Doublet Detection in Single-Cell RNA Sequencing Data Using Artificial Nearest Neighbors. *Cell Syst.* 2019;8:329-337.e4.

REVIEWER COMMENTS

Reviewer #1 (Remarks to the Author):

1. There are still numerous grammatical errors throughout the manuscript.
2. Line 372 – I believe they meant to say PDGF induced SMC de-differentiation.
3. Authors ignored my recommendation to analyze BCA atherosclerotic lesions which much more closely resemble human lesions. Indeed, the most important studies in the paper focus nearly exclusively on analysis of Extracellular Traps derived from SMC-derived macrophage-like cells within aortic root lesions. However, humans do not develop lesions at this site and mouse aortic root lesions poorly resemble human lesions morphologically. The authors probably did not harvest BCA lesions and thus would have to repeat their entire study to comply. An alternative would be for them to provide some evidence the processes they describe in mouse models of atherosclerosis also occur in human lesions.
4. The Wirka et al paper cited (REF 51) failed to resolve many SMC-derived phenotypes including macrophage-like cells highly relevant to the present studies likely due to methodological issues. They also incorrectly concluded that SMC-derived cells only have beneficial roles in lesions a finding completely at odds with the present studies. Authors should cite papers by Low et al. and Alencar et al 2021 Circulation which are consistent with the results of the present studies.
5. The font size on multiple figures is so small they are unreadable (e.g. Figures 5c, S4a, S5c, e, g).

Reviewer #2 (Remarks to the Author):

You sufficiently addressed my concerns.

by Martin Herrmann

A point-by-point response to the Reviewers' critique

We sincerely thank the three reviewers for the valuable and constructive suggestions on our manuscript. Please find our point-by-point response below. All changes and revisions in the edited manuscript are highlighted in yellow text color for tracking purposes. New figure panels and supplementary figures are indicated by yellow letters and yellow-color figure legends.

Reviewer #1:

Response: We thank you very much again for this feedback and your valuable critique in the review.

1. There are still numerous grammatical errors throughout the manuscript.

Response: We feel sorry for that. We have corrected the grammatical errors and the revised parts are marked in yellow.

2. Line 372 – I believe they meant to say PDGF induced SMC de-differentiation.

Response: Thanks for your valuable suggestion. We have corrected this in line 370.

3. Authors ignored my recommendation to analyze BCA atherosclerotic lesions which much more closely resemble human lesions. Indeed, the most important studies in the paper focus nearly exclusively on analysis of Extracellular Traps derived from SMC-derived macrophage-like cells within aortic root lesions. However, humans do not develop lesions at this site and mouse aortic root lesions poorly resemble human lesions morphologically. The authors probably did not harvest BCA lesions and thus would have to repeat their entire study to comply. An alternative would be for them to provide some evidence the processes they describe in mouse models of atherosclerosis also occur in human lesions.

Response: Thanks for bringing up this suggestion. We apologize for ignoring the value of atherosclerosis plaque within BCA, which more resembles human lesions morphologically. Following the Reviewer's suggestions, we verify our results again in BCA lesions harvested before from HFD-fed *Myh11^{Cre}Pad4^{fllox/fllox}* mice and *Pad4^{fllox/fllox}* mice. By using the IF test, we find that the inhibition of PAD4 in VSMCs indeed suppresses the ETs released from CD68⁺ VSMCs in BCA lesions of *Myh11^{Cre}Pad4^{fllox/fllox}* mice, which is shown in new **Fig 3H & 3J-L**. Histologically, inhibition of CD68⁺ VSMCs generated ETs indeed improves the stability of the atherosclerosis lesions of BCA with a reduction of necrotic areas, an increase of fibrous areas, and a reduction of MMP9-positive or CD68-positive areas. The verified results have been presented in new **Fig 3M-3T**. And the expression of SOCS1 and GSDMD is also downregulated in BCA lesions from *Myh11^{Cre}Pad4^{fllox/fllox}* mice (new **Fig 5L-O**). Meanwhile, we also analyze the m-IF results of BCA lesions from HFD-fed *B6-G/R Myh11^{Cre}Pad4^{fllox/fllox}* and *B6-G/R Myh11^{Cre}* mice to verify that the ETs from CD68⁺ VSMCs are diminished after PAD4 knock out within VSMCs (shown in new **Fig 4E-**

4G). Furthermore, we also find ETs are generated from CD68⁺ VSMCs in three specimens from human aspiration atherosclerosis plaque and find that areas of ETs positive CD68⁺ VSMCs are related to MMP9 positive areas in a series adjacent section, which is consistent with our finding results that ETs generated from CD68⁺ VSMCs induced plaque's instability in mice (shows in new Supplementary Fig 4O).

Fig 3H & 3J-L :

Fig 3M-3T :

Fig 5L-O :

Fig 4E-4G:

Supplementary Fig 4O :

4. The Wirka et al paper cited (REF 51) failed to resolve many SMC-derived phenotypes including macrophage-like cells highly relevant to the present studies likely due to methodological issues. They also incorrectly concluded that SMC-derived cells only have beneficial roles in lesions a finding completely at odds with the present studies. Authors should cite papers by Low et al. and Alencar et al 2021 Circulation which are consistent with the results of the present studies.

Response: We thank you for raising this point. According to the Reviewer's suggestion, we have deleted the wrong reference paper in REF 51 and cited papers by Alencar et al. 2021 Circulation in the new REF 51.

5. The font size on multiple figures is so small they are unreadable (e.g. Figures 5c, S4a, S5c, e, g).

Response: We thank the Reviewer's suggestion. We have increased the font size in the new image of our revised manuscript.

Reviewer #2 (Remarks to the Author):

You sufficiently addressed my concerns.

Response: Thank you very much again for this feedback and for your valuable critique of our initial manuscript version.

Reviewer #3

Response: Thank you very much again for your valuable critique.

REVIEWERS' COMMENTS

Reviewer #1 (Remarks to the Author):

Authors have addressed my previous concerns.

A point-by-point response to the Reviewers' critique

We sincerely thank the three reviewers for the valuable and constructive suggestions on our manuscript.

Reviewer #1: Authors have addressed my previous concerns.

Response: Thank you very much again for this feedback and for your valuable critique of our initial manuscript version.